# Ubiquitin-dependent regulation of a conserved DMRT protein controls sexually dimorphic synaptic connectivity and behavior

Emily A Bayer[1], Rebecca C Stecky[1], Lauren Neal[1], Phinikoula S Katsamba[2], Goran Ahlsen[2], Vishnu Balaji[3], Thorsten Hoppe[3,4], Lawrence Shapiro[2], Meital Oren-Suissa[5], Oliver Hobert[1,2]*

[1]Department of Biological Sciences, Howard Hughes Medical Institute, Columbia University, New York, United States; [2]Department of Biochemistry and Molecular Biophysics, Columbia University Irving Medical Center, New York, United States; [3]Institute for Genetics and Cologne Excellence Cluster on Cellular Stress Responses in Aging-Associated Diseases (CECAD), University of Cologne, Cologne, Germany; [4]Center for Molecular Medicine Cologne (CMMC), University of Cologne, Cologne, Germany; [5]Weizmann Institute of Science, Department of Neurobiology, Rehovot, Israel

**Abstract** Sex-specific synaptic connectivity is beginning to emerge as a remarkable, but little explored feature of animal brains. We describe here a novel mechanism that promotes sexually dimorphic neuronal function and synaptic connectivity in the nervous system of the nematode *Caenorhabditis elegans*. We demonstrate that a phylogenetically conserved, but previously uncharacterized Doublesex/Mab-3 related transcription factor (DMRT), *dmd-4*, is expressed in two classes of sex-shared phasmid neurons specifically in hermaphrodites but not in males. We find *dmd-4* to promote hermaphrodite-specific synaptic connectivity and neuronal function of phasmid sensory neurons. Sex-specificity of DMD-4 function is conferred by a novel mode of posttranslational regulation that involves sex-specific protein stabilization through ubiquitin binding to a phylogenetically conserved but previously unstudied protein domain, the DMA domain. A human DMRT homolog of DMD-4 is controlled in a similar manner, indicating that our findings may have implications for the control of sexual differentiation in other animals as well.

*For correspondence:
or38@columbia.edu

## Introduction

Across both invertebrate and vertebrate phyla, adult animals usually display striking sexually dimorphic features (*Fairbairn, 2013*). These features range from morphological secondary sex characteristics such as pigmentation patterns to complex behavioral programs executed by one sex but not the other. Many sexually dimorphic morphologies and behaviors arise during specific phases of juvenile development, indicating the existence of developmental timing mechanisms that must intersect with the genetic programs that control sexual identity to produce sexually dimorphic features and structures of the adult animal (*Faunes and Larraín, 2016*; *Tena-Sempere, 2013*).

The existence of sexually dimorphic behaviors suggests the existence of sexually dimorphic features in the underlying neural circuits that govern such behaviors. The recently reconstructed synaptic wiring diagram of an adult *C. elegans* male provides novel vistas on the sex-specificity of nervous system anatomy, revealing the existence of sex-specific synaptic connections between neurons

present in both sexes ('sex-shared neurons')(*Cook et al., 2019*; *Jarrell et al., 2012*; *White et al., 1986*). Moreover, specific sexually dimorphic circuits comprised of sex-shared neurons have been shown to generate behavioral dimorphisms in *C. elegans* (*Barr et al., 2018*; *Emmons, 2018*; *Oren-Suissa et al., 2016*; *Ryan et al., 2014*; *White et al., 2007*). How are sex-shared neurons genetically instructed to undergo sexually dimorphic maturation that results in changes in sex-specific synaptic connectivity and behavior?

While sexual differentiation is a vital part of development across a breadth of species, the upstream components of genetic pathways known to trigger sex determination are strikingly divergent. Multiple distinct primary sex determination mechanisms have arisen at various points in evolutionary history (autosomal sex determination, male or female heterogamy, fertilization status, or environmental cues), all with the ultimate downstream goal of producing stable mating types (*Fairbairn, 2013*). Nevertheless, in many metazoans, these divergent modes of sex determination employ, at some step of the sexual differentiation process, member(s) of the DMRT (Doublesex and MAB-3 related transcription factor) family of transcription factors (*Kopp, 2012*; *Matson and Zarkower, 2012*). As their name implies, DMRT family members were first identified in *D. melanogaster* (Doublesex) and *C. elegans* (MAB-3) based on their roles in directing sexually dimorphic development (*Baker and Ridge, 1980*; *Shen and Hodgkin, 1988*). DMRT family members were subsequently found to also control sexual differentiation of gonadal structures in vertebrates (*Kopp, 2012*; *Matson and Zarkower, 2012*; *Raymond et al., 1998*).

DMRT-encoding genes evolved from three ancestral clusters of genes that were present in the ancestor of bilateria and radiated independently in multiple distinct lineages unique to each phylum (*Mawaribuchi et al., 2019*; *Volff et al., 2003*; *Wexler et al., 2014*). Those DMRT proteins most extensively characterized in controlling sex-specific differentiation in the brain – *mab-3*, *mab-23*, and *dmd-3* in *C. elegans* and *doublesex* in *Drosophila* (*Lints and Emmons, 2002*; *Mason et al., 2008*; *Rideout et al., 2010*; *Siehr et al., 2011*; *Taylor et al., 1994*; *Verhulst and van de Zande, 2015*; *Yi et al., 2000*) – are examples of such radiated DMRTs. Since those DMRTs have no clear orthologs outside their respective phyla, the question arises to what extent the phylogenetically more deeply conserved DMRT proteins are involved in sexual differentiation in the brain.

We describe here the function of a previously uncharacterized DMRT gene, *dmd-4,* that is deeply conserved across phylogeny. We show that *dmd-4* is expressed in a specific set of neurons and in a subset of sensory neurons in a sex-specific manner. We show that DMD-4 is required to generate sexually dimorphic connectivity and behavioral output of sex-shared neurons. Unexpectedly, we found that sexually dimorphic expression of DMD-4 is established post-translationally via a previously uncharacterized, phylogenetically conserved domain, the DMA domain, which we identify as a ubiquitin-binding domain both in *C. elegans* and *H. sapiens*. Ubiquitin binding appears to be critical to stabilize DMD-4 protein in all cell types, but this ubiquitin-mediated stabilization is dynamically controlled in a cell-, time- and sex-specific manner to sculpt sexually dimorphic expression of DMD-4 in phasmid neurons during sexual maturation.

## Results

### Expression pattern of *dmd-4*

DMD-4 is one of ten DMRT proteins encoded by the *C. elegans* genome, only two of which, DMD-4 and DMD-5, are phylogenetically conserved in vertebrates (*Volff et al., 2003*; *Wexler et al., 2014*). Neither the expression nor the function of DMD-4 have previously been described. To determine the expression pattern of *dmd-4,* we used CRISPR/Cas9 genome engineering to tag the endogenous *dmd-4* locus at the 3' end of its coding region with *gfp*. We examined expression in sexually mature, adult animals and identified two sites of sexually dimorphic expression in the tail of the animal. DMD-4::GFP is expressed in PHA and PHB phasmid sensory neuron pairs of adult hermaphrodites, but not adult males (*Figure 1A,B*; *Figure 1—figure supplement 1*). Electron micrographic reconstruction of the synaptic connectivity of males and hermaphrodites revealed that these neurons display highly dimorphic synaptic connectivity (*Cook et al., 2019*; *Jarrell et al., 2012*; *Figure 1C*).

In addition to this sexually dimorphic expression, we observed sexually non-dimorphic expression of DMD-4 in a subset of head sensory neurons (AFD, AWB, AWC, ASE, ASG, ASH, BAG), a single pharyngeal neuron (I5) and a number of MS blastomere derived cells - the head mesodermal cell

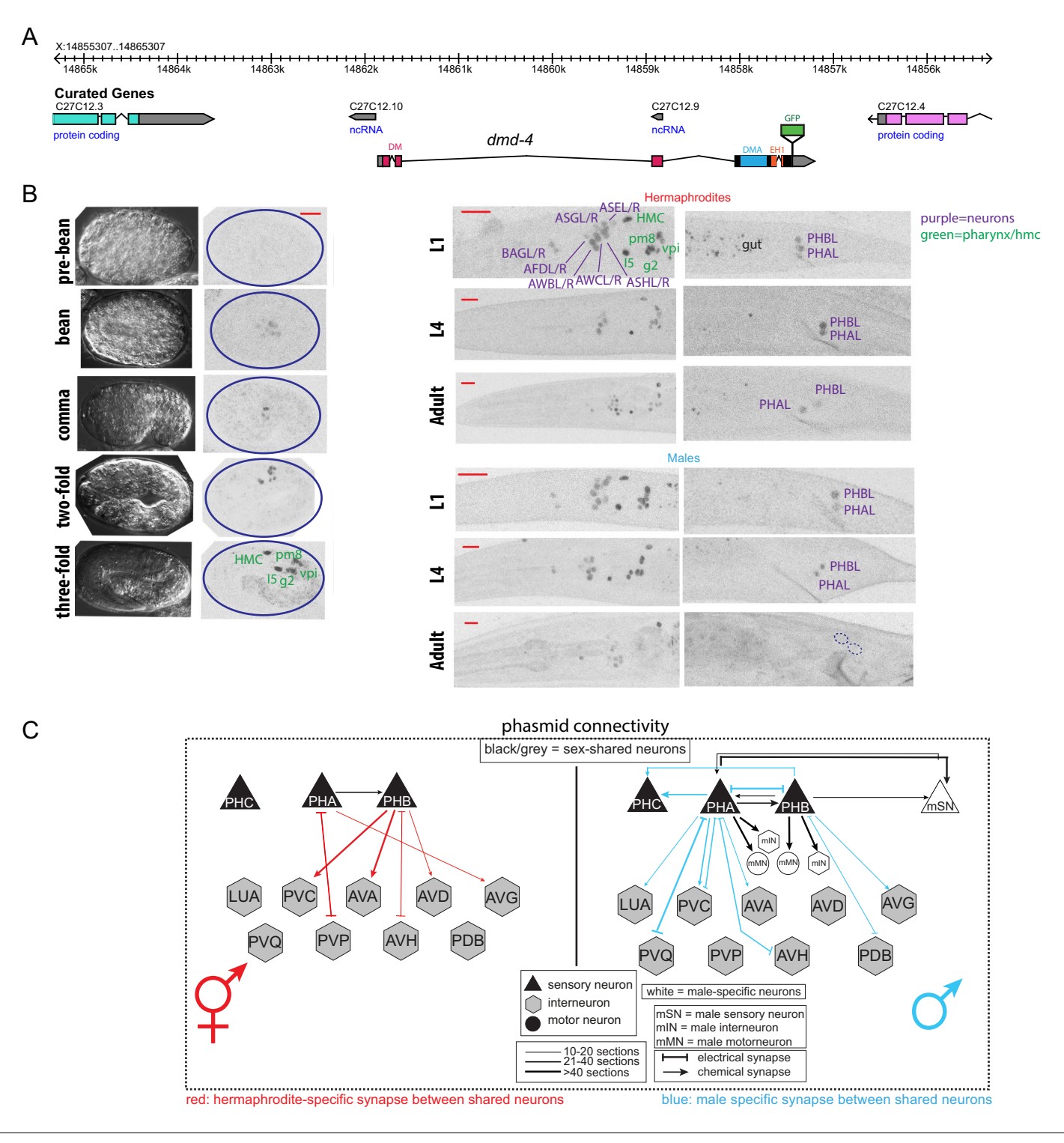

**Figure 1.** DMD-4 shows both sexually dimorphic and non-dimorphic expression. (**A**) Schematic of the GFP-tagged *dmd-4* locus in the *dmd-4 (ot935)* allele. The DM, DMA, and EH1 domains of DMD-4 are indicated in the colored boxes and labeled above the locus. (**B**) DMD-4::GFP is expressed in head sensory neurons and pharyngeal cells/the head mesodermal cell (hmc) in both sexes, and the phasmid sensory neurons PHA and PHB in both sexes until adulthood, when it is degraded in the male phasmids. Expression in the pharyngeal cells onsets during the 'bean' stage of embryogenesis. GFP is shown as color-inverted black and white. All scale bars (red horizontal bars) indicate 10 microns. Head images are maximum intensity projections of the entire worm; tail images are maximum intensity images of one half of the tail to clearly display phasmid nuclei. (**C**) Sexually dimorphic connectivity of the PHA and PHB neurons based on electron micrograph reconstruction (*Cook et al., 2019*; *Jarrell et al., 2012*).

*Figure 1 continued on next page*

Figure 1 continued

The online version of this article includes the following figure supplement(s) for figure 1:

**Figure supplement 1.** DMD-4::GFP is dimorphically expressed in the phasmid neurons.

(hmc), a single pharyngeal muscle (pm8), a pharyngeal gland cell pair (g2L/R) and four of the six pharyngeal intestinal valve cells (*Figure 1B*).

The onset of sexual differentiation in the *C. elegans* nervous system occurs at different stages of embryonic and postembryonic development (*Barr et al., 2018*; *Emmons, 2005*; *Sulston and Horvitz, 1977*; *Sulston et al., 1983*). To address the timing of sexually dimorphic DMD-4 expression, we examined expression of the *gfp*-tagged *dmd-4* locus throughout all developmental stages. DMD-4::GFP fluorescence is first detectable in mid-embryonic development, with a pattern that is restricted to all the pharyngeal and the hmc cells described above. Onset of expression in the non-pharyngeal nervous system is first observed at around hatching in the same set of head and tail sensory neurons as observed in the adult stage, including the phasmid neurons PHA and PHB. During larval development, *dmd-4::gfp* expression is the same in both sexes until the L4 to adult molt when DMD-4::GFP protein starts to disappear from PHA and PHB in the male, but not the hermaphrodite (*Figure 1—figure supplement 1*). We conclude that sexually dimorphic *dmd-4::gfp* expression in the adult PHA and PHB neurons is the result of male-specific downregulation of *dmd-4::gfp* from these neurons. These dynamics contrast with the ontogeny of other sexually dimorphically expressed genes in the *C. elegans* nervous system, which often are not expressed at all until sexual maturation and then turn on in a sex-specific manner (e.g. *daf-7* in ASJ, *flp-13, eat-4* in PHC, *srj-54* in AIM) (*Hilbert and Kim, 2017*; *Lawson et al., 2019*; *Serrano-Saiz et al., 2017*). Moreover, the hermaphrodite-specificity of DMD-4 expression also contrasts with the strict male-specificity of other DMRT proteins known to be dimorphically expressed in *C. elegans* (*Lints and Emmons, 2002*; *Mason et al., 2008*; *Oren-Suissa et al., 2016*; *Yi et al., 2000*).

## Generation of a nervous system-specific knock-out allele of *dmd-4*

To assess *dmd-4* function in the nervous system and, specifically, in the phasmid neurons, we used CRISPR/Cas9 genome engineering to delete the first two exons of the gene which code for most of the DNA binding domain (*Figure 2A*). This putative null allele (*ot933*) results in highly penetrant embryonic lethality (*Figure 2A*). A deletion allele generated by the *C. elegans* knockout consortium (*tm1951*), that eliminates parts of the DM domain, also results in lethality. Null mutant animals that escape embryonic lethality are scrawny and display a stuffed anterior intestine (*Figure 2—figure supplement 1A,B*). The stuffed anterior intestine may relate to developmental defects of the hmc, which fail to express a cell fate marker (*arg-1::gfp*) in *dmd-4* mutants (*Figure 2—figure supplement 1C*). Another deletion allele that we generated using CRISPR/Cas9 leaves the DNA binding domain unaffected, but deletes other domains of the protein (DMA domain and EH1 domain)(*Figure 2A*), but this allele (*ot896)* results in no embryonic lethality.

We sought to generate a nervous system-specific allele of the *dmd-4* locus that we would expect to be viable and thus allow us to study the role of *dmd-4* in the sexually dimorphic phasmid sensory neurons. To this end, we aimed to identify (and then delete) the *cis*-regulatory element required for *dmd-4* expression in the nervous system. Through the fusion of individual segments of the *dmd-4* locus to *gfp*, we identified the third intron of the *dmd-4* locus as containing the *cis*-regulatory information for all nervous system expression (*Figure 2B*). Using CRISPR/Cas9 genome engineering, we deleted this enhancer in the context of the *dmd-4::gfp* allele (*dmd-4(ot957ot935))*, and indeed found that in these animals *dmd-4::gfp* expression was lost in all head and tail neurons at all developmental stages, but unaffected in the pharynx and hmc (*Figure 2C*). Most importantly, *dmd-4(ot957ot935)* mutants showed none of the lethality or intestinal stuffing phenotypes associated with removal of the gene from all tissues (*Figure 2—figure supplement 1A,B*), allowing us to characterize the function of *dmd-4* in the nervous system.

Using this nervous system-specific allele, we evaluated several markers of neuronal identity to assess whether neurons that express *dmd-4* were properly specified. We found no evidence of defects in general neuronal specification or morphology; all head sensory neurons as well as the phasmid neurons were generated in *dmd-4(ot957ot935)* mutant hermaphrodites and had a normal

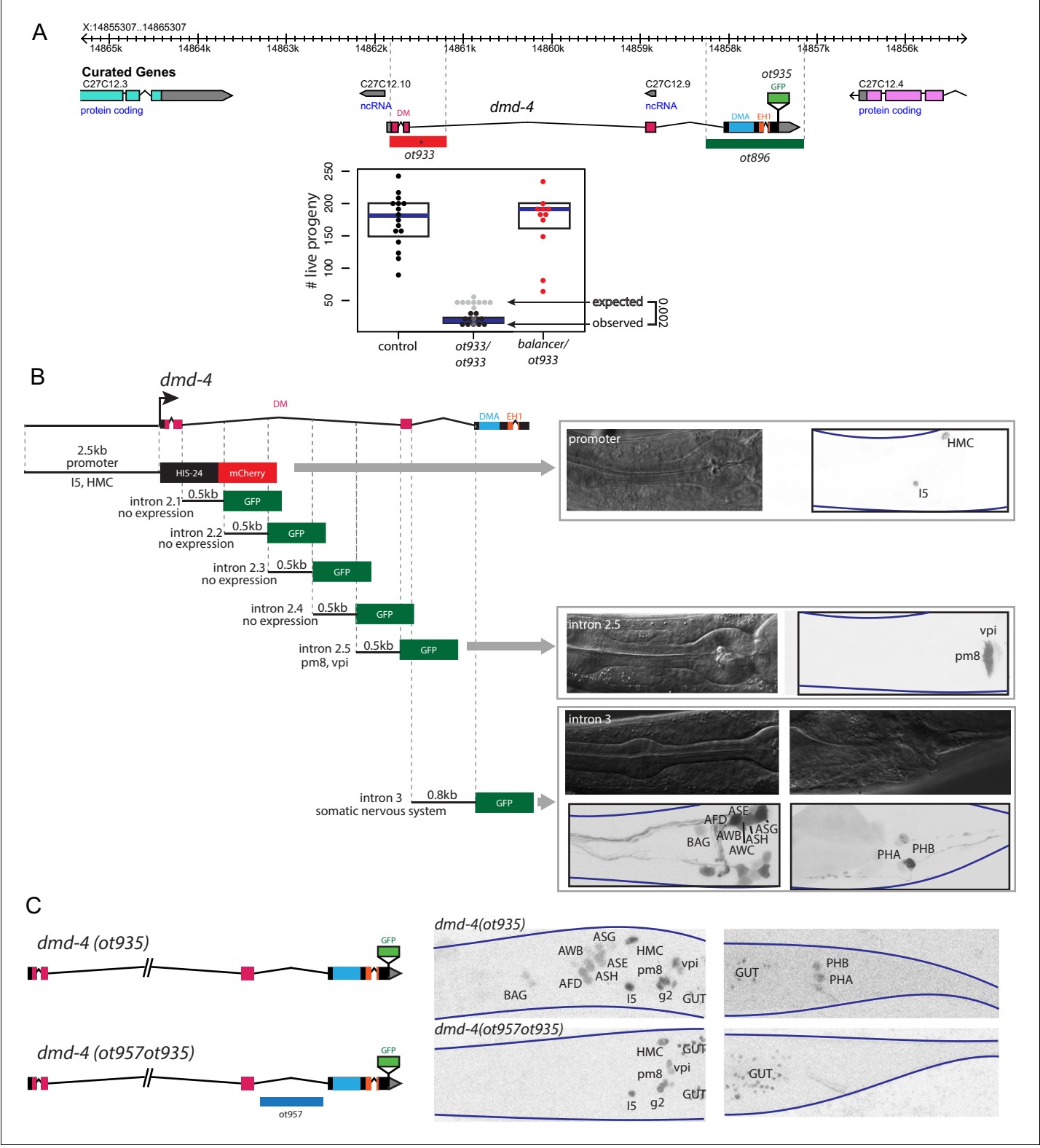

**Figure 2.** Generation of a viable *cis*-regulatory allele of *dmd-4*. (**A**) The *ot933* probable null allele of *dmd-4* results in embryonic lethality and severe feeding defects in mutants that survive to adulthood. Deletion alleles generated by CRISPR/Cas9 are shown in the context of the *dmd-4* locus above, the penetrance of *ot933* embryonic lethality was quantified below by comparing the number of expected *ot933* homozygous progeny from a balanced heterozygote mother (25% of total brood) with observed *ot933* homozygous progeny. The difference between the observed number of *ot933* homozygous progeny and expected homozygous progeny gives a measure of the penetrance of embryonic lethality. We observe a mean of 52%

*Figure 2 continued on next page*

*Figure 2 continued*

embryonic lethality, and comparing observed to expected *ot933* progeny counts is significant by Wilcoxon rank-sum test. Each dot represents the brood size of one hermaphrodite, with blue bars indicating median and black boxes indicating quartiles. We show both observed and expected numbers of *ot933* progeny in the figure panel. (B) The enhancers for I5/hmc expression, pharyngeal muscle expression, and somatic nervous system expression are located in separable regions of the *dmd-4* promoter and introns. All constructs we examined are schematized, those that showed expression (promoter, intron 2.5, and intron 3) are shown in adult hermaphrodites to the right. GFP and mCherry are shown as color-inverted black and white, boundaries of the heads and tails of animals are marked with navy lines in all panels. Expressing cells are labeled on the GFP/mCherry images. (C) Generation of the *dmd-4 (ot957ot935)* nervous system null allele. Deletion of the third intron is schematized in the context of the *dmd-4* locus above, color-inverted black and white GFP images of *dmd-4 (ot935[*DMD-4::GFP]) and *dmd-4 (ot957ot935)* hermaphrodites are shown below.

The online version of this article includes the following figure supplement(s) for figure 2:

**Figure supplement 1.** DMD-4 expression in the pharynx is required for proper anterior intestinal function.
**Figure supplement 2.** General neuron identity is unaffected in *dmd-4 (ot957ot935)* mutants.

appearance of axon and dendrite morphologies (*Figure 2—figure supplement 2*). Furthermore, we examined transgenic reporters for neurotransmitter identity, neuropeptides, and receptors, and found no loss or misexpression of these genes in the amphid and phasmid neurons of *dmd-4 (ot957ot935)* mutant animals (*Figure 2—figure supplement 2*). Lastly, behaviors associated with the function of head amphid sensory neurons also appear unaffected (*Figure 2—figure supplement 2*).

## *dmd-4* mutants display defects in sexually dimorphic synaptic connectivity

The sex-shared phasmid sensory neurons are notable for their strikingly dimorphic synaptic connectivity patterns (*Cook et al., 2019*; *Jarrell et al., 2012*; *Figure 1C*). Not only do both the PHA and PHB neurons generate synaptic connections to male-specific neurons, but they also display dimorphic synaptic connections to neurons that are shared between the two sexes (*Figure 1C*). We have previously used in vivo synapse labeling techniques, GRASP (GFP-Reconstitution Across Synaptic Partners)(*Feinberg et al., 2008*) and iBLINC (*Desbois et al., 2015*), to show that juvenile stages exhibit non-dimorphic 'sex-hybrid' connectivity and that during sexual maturation, some synapses are then pruned, while others are maintained (*Figure 3A*; *Oren-Suissa et al., 2016*). Unlike many pruning/maintenance events during nervous system development in other animal species (*Katz and Callaway, 1992*; *Katz and Shatz, 1996*; *Morgan et al., 2011*; *Vonhoff and Keshishian, 2017*), this synaptic maintenance is not dependent on sex-specific patterns of neuronal activity. We arrived at this conclusion by genetic silencing of the phasmid neurons or reducing synaptic signaling in *unc-13* mutant animals, neither of which affects sexually dimorphic synapse pruning/maintenance (*Figure 3— figure supplement 1*).

Using these GRASP reagents we assessed a subset of the hermaphrodite-specific and male-specific connections in the adult phasmid neurons of *dmd-4(ot957ot935)* mutant animals. We observed selective defects in sexually dimorphic synaptic connectivity. While hermaphrodite-specific synaptic connections (between PHA>AVG and PHB>AVA) are unaffected in *dmd-4* mutants (*Figure 3A,B*, *Figure 3—figure supplement 2*), male-specific synapses (PHB>AVG) failed to be pruned in hermaphrodite *dmd-4* mutants (*Figure 3A,C*, *Figure 3—figure supplement 2*; *Oren-Suissa et al., 2016*). We also developed a PHA-specific transgene using a synaptic vesicle-associated protein, RAB-3, to visualize all presynaptic zones of PHA as discrete BFP puncta (*Figure 3C*). Transgenic animals show the predicted sexual dimorphism based on electron micrographic reconstruction of PHA synaptic connectivity, with wild-type males displaying more synaptic outputs than wild-type hermaphrodites (*Figure 1C*, *Figure 3C*, *Figure 3—figure supplement 2*; *Cook et al., 2019*; *Jarrell et al., 2012*). In *dmd-4(ot957ot935)* mutant animals this sexual dimorphism is also lost, corroborating the notion that synapses fail to be pruned (*Figure 3C*, *Figure 3—figure supplement 2*).

We rescued the PHB>AVG synaptic pruning defect by expressing *dmd-4* cDNA under the control of a PHB-specific promoter, confirming the cell-autonomous role of *dmd-4* in the presynaptic neuron to control sex-specific synaptic pruning (*Figure 3C*, *Figure 3—figure supplement 2*). Taken together our results indicate that *dmd-4* functions in the phasmid neurons to promote pruning of phasmid synaptic connections in hermaphrodite animals during sexual maturation.

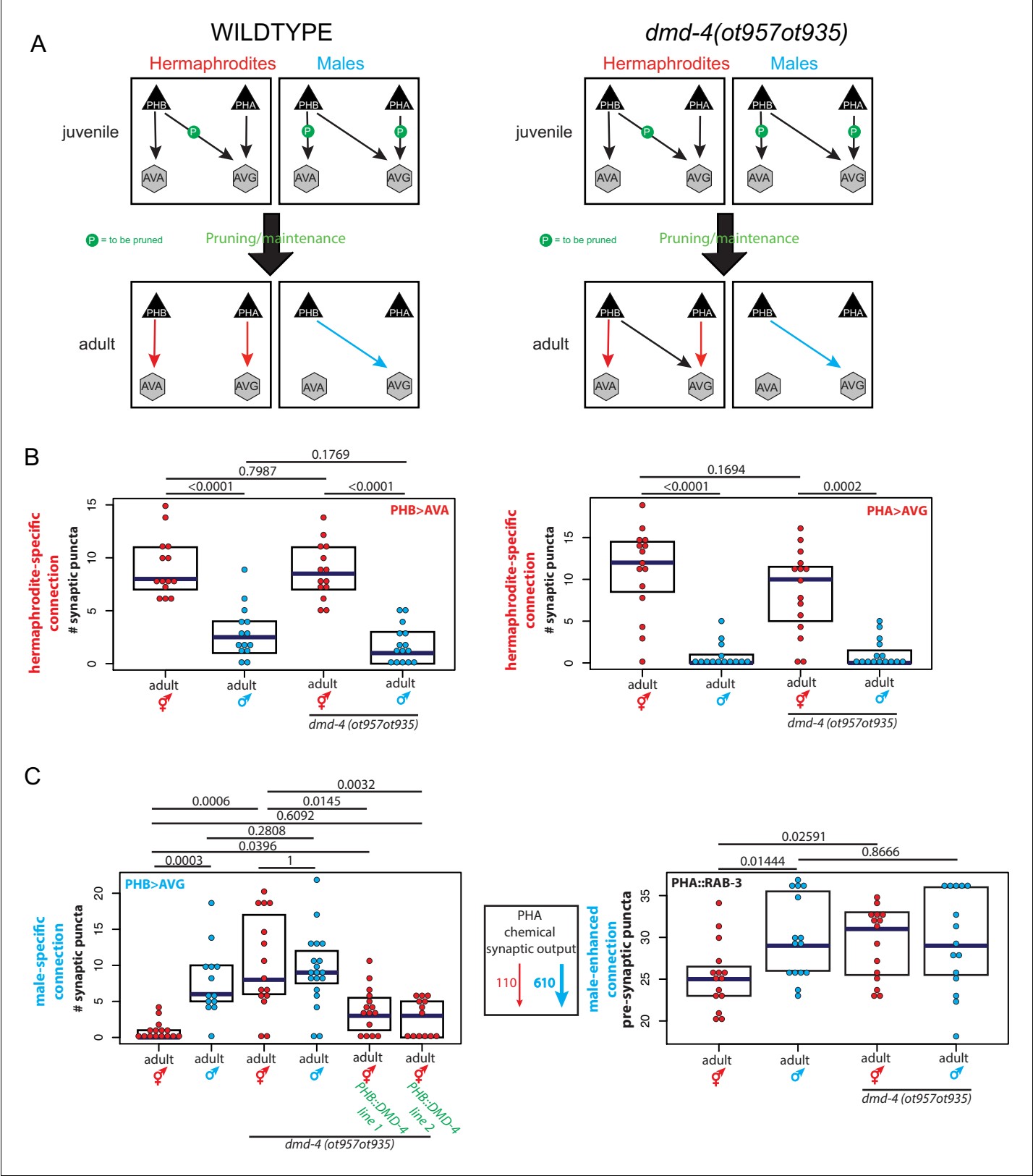

**Figure 3.** *dmd-4* is required for sexually dimorphic synaptic connectivity. (**A**) Sexually dimorphic connectivity of the phasmid neurons in wild-type animals and *dmd-4 (ot957ot935)* animals (quantified in B, representative images shown in *Figure 3—figure supplement 2*) is schematized. Adult sexual dimorphism is generated by synaptic pruning of juvenile 'sex-hybrid' connectivity (*Oren-Suissa et al., 2016*). (**B**) The hermaphrodite-specific

*Figure 3 continued on next page*

*Figure 3 continued*

connections between PHB>AVA and PHA>AVG are unaffected in *dmd-4 (ot957ot935)* mutants. Each dot represents one animal (red = hermaphrodite, cyan = male, in all figures), blue bars show median, black boxes represent quartiles. *P* values are shown above horizontal black bars by Wilcoxon rank-sum test with Bonferroni corrections for multiple testing where applicable (see Materials and methods). Representative images are shown in *Figure 3—figure supplement 2*. (C) The male-specific PHB>AVG connection is not pruned in *dmd-4 (ot957ot935)* mutant hermaphrodites, and this phenotype can be rescued by expressing *dmd-4* cDNA specifically in the PHB neurons ('PHB::DMD-4'). We quantified two independent transgenic lines for mutant rescue. *dmd-4 (ot957ot935)* mutant hermaphrodites also show an increased number of BFP puncta in the BFP::RAB-3 construct labeling pre-synaptic vesicles. The sexual dimorphism of PHA chemical synaptic output is shown on the left, based on the number of EM sections where PHA is presynaptic in both sexes, 110 sections in hermaphrodites and 610 sections in males, (*Cook et al., 2019*). We term this connectivity 'male-enhanced' as it is present in both sexes but stronger in the male. Representative images are shown in *Figure 3—figure supplement 2*.

The online version of this article includes the following figure supplement(s) for figure 3:

**Figure supplement 1.** Phasmid synaptic pruning is not activity-dependent.

**Figure supplement 2.** *dmd-4* is required for dimorphic synaptic connectivity.

## Behavioral defects of *dmd-4* mutants

Sexually dimorphic connectivity of the phasmid neurons gives rise to sexually dimorphic behavioral outputs in adult animals. In larval animals, the phasmid neurons function to mediate chemosensory avoidance behavior (*Hilliard et al., 2002*; *Zou et al., 2017*), a functional output which is maintained in adult hermaphrodites, but lost in adult males (*Oren-Suissa et al., 2016*). *dmd-4* mutants exhibit no defects in SDS avoidance in juvenile stages, before sexual maturation (*Figure 4A*). However, as adults, these animals display a partial masculinization of SDS avoidance behavior, such that mutant hermaphrodites were less responsive to the noxious stimulus, although still intermediate relative to control adult males (*Figure 4A*). It is possible that this defect is the result of lack of proper pruning of synapses, but it could also be the result of other functions of *dmd-4* in the phasmid neurons.

## Control of spatiotemporal and sexual specificity of DMD-4 expression

We sought to place the regulation of DMD-4 expression in the context of other regulatory pathways that control spatial, sexual and temporal aspects of gene expression in the nervous system. First, to ask how *dmd-4* expression is established specifically in the phasmid neurons in an initially non-sex-specific manner at hatching, we turned to terminal selector type transcription factors, master regulatory factors that control multiple aspect of a neuron's identity (*Hobert, 2016*). Based on previous mutant analysis, the *ceh-14* LIM homeobox transcription factor is a terminal selector for the phasmid neurons (*Kagoshima et al., 2013*; *Serrano-Saiz et al., 2013*). We find that in *ceh-14* null mutants, *dmd-4::gfp* expression is selectively lost in the phasmid neurons in both sexes during larval stages and in the adult stage in hermaphrodites (*Figure 5A*). This is also consistent with the observation that *dmd-4(ot957ot935)* mutant hermaphrodites display masculinization of SDS avoidance behavior (*Figure 4A*), a phenotype observed in *ceh-14* mutant hermaphrodites (*Oren-Suissa et al., 2016*).

We found that the well-characterized heterochronic gene regulatory pathway (*Ambros and Horvitz, 1984*; *Rougvie and Moss, 2013*) affects the timing of the downregulation of DMD-4::GFP in male phasmid neurons at the adult stage. During the third larval stage, the heterochronic regulator LIN-41, a RNA-binding protein, is down-regulated to allow progression into the final larval stage and then adulthood (*Slack et al., 2000*), which is one of several developmental time windows during which sexual maturation events occur in several distinct parts of the nervous system (*Lawson et al., 2019*; *Pereira et al., 2019*). We generated transgenic animals that constitutively express LIN-41 in the nervous system and asked whether ectopically maintaining LIN-41 throughout all larval stages to adulthood is sufficient to maintain DMD-4 expression in the adult male phasmids. Indeed, we found that DMD-4 was ectopically maintained in these transgenic male animals (*Figure 5B*), demonstrating that the heterochronic pathway functions to regulate the timing of DMD-4 degradation.

To address how the sexual specificity of *dmd-4::gfp* downregulation is controlled, we turned to the TRA-1 master regulatory factor of sex determination, a transcription factor expressed in somatic cells of the hermaphrodite but degraded in males (*Schvarzstein and Spence, 2006*; *Zarkower and Hodgkin, 1992*). Manipulation of TRA-1 activity has been previously found to be required for the establishment of a number of sexually dimorphic gene expression programs in the nervous system, as well as sexually dimorphic synaptic connectivity (*Lee and Portman, 2007*; *Mowrey et al., 2014*; *Oren-Suissa et al., 2016*; *White et al., 2007*). We found that genetic removal of *tra-1*, in *tra-1*

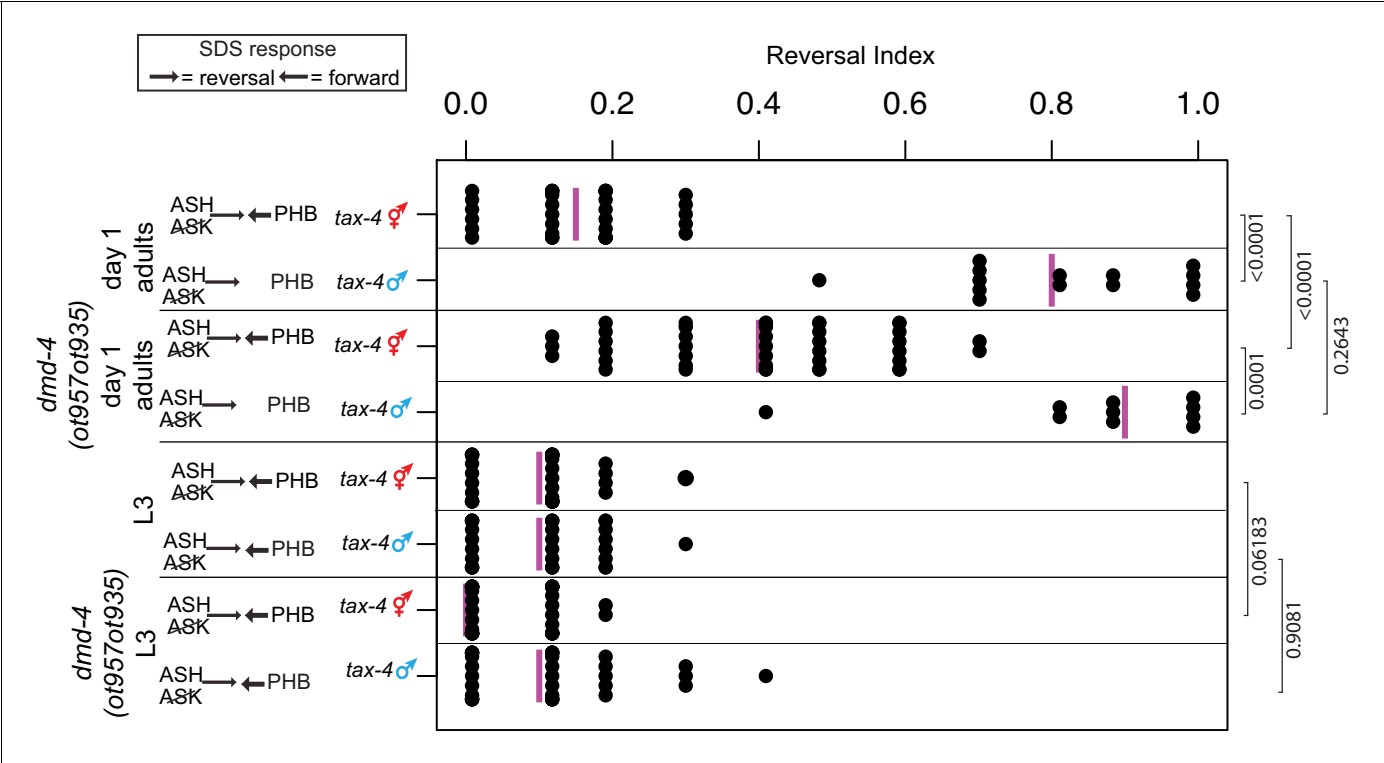

**Figure 4.** *dmd-4* affects sexually dimorphic behavior. Noxious chemosensory avoidance is mediated by the PHB neuron (*Hilliard et al., 2002*), and sexually dimorphic in adults (*Oren-Suissa et al., 2016*). Left panel shows predicted synaptic input into avoidance behavior by relevant amphid and phasmid neurons. All experiments were done using *tax-4* (*p678*) mutants to disable amphid input and uncover phasmid function, as previously described (*Hilliard et al., 2002*). *dmd-4* (*ot957ot935*) adult hermaphrodites show reduced chemosensory avoidance behavior, but this is not evident in larval animals. Each dot represents the average reversal index of one animal over 10 experimental trials, median shown with vertical magenta bar. *P* values shown to the right by Wilcoxon rank-sum test with Bonferroni corrections for multiple testing (where applicable; see Materials and methods).

(*e1099*) XX pseudomales, leads to a downregulation of DMD-4 expression in hermaphrodite phasmid neurons at the adult stage (*Figure 5C*). Conversely, force-expression of an engineered, activated version of the TRA-1 protein which escapes protein degradation (*Schvarzstein and Spence, 2006*) in the phasmid neurons of both sexes results in ectopic DMD-4 expression in male phasmid neurons (*Figure 5D*). We conclude that the transcriptional regulator TRA-1 normally acts to promote DMD-4::GFP expression in hermaphrodite phasmid neurons upon sexual maturation, while its absence in adult male phasmids results in DMD-4 loss.

## DMD-4 downregulation is controlled post-transcriptionally

Our implication of TRA-1, a transcription factor, in maintaining DMD-4 expression in hermaphrodite, but not male phasmid neurons (where TRA-1 is not expressed) suggested a transcriptional control mechanism for DMD-4 downregulation. However, in the context of our analysis of the *cis*-regulatory control regions of *dmd-4* expression, we noted that the transcriptional reporter containing the nervous system enhancer for *dmd-4* expression (*Figure 2B*), was expressed in the phasmid neurons of both adult hermaphrodite and adult male animals (*Figure 6A*). This suggested that the regulatory information generating DMD-4 sexual dimorphism is not encoded on the transcriptional level. To define the potential mechanism of posttranscriptional regulation, we expressed *gfp* tagged *dmd-4* cDNA with a heterologous 3′ UTR under control of a non-dimorphic PHA-specific and PHB-specific promoter. In transgenic animals, these constructs show dimorphic GFP fluorescence in the phasmid neurons, identical to the endogenous DMD-4::GFP expression pattern (*Figure 6B*). Since the *dmd-4* cDNA alone confers male-specific downregulation, sexually dimorphic DMD-4 presence may be controlled on the level of stability/degradation of the protein. In the next section, we describe experiments that demonstrate that a specific domain and amino acids within the DMD-4 protein can

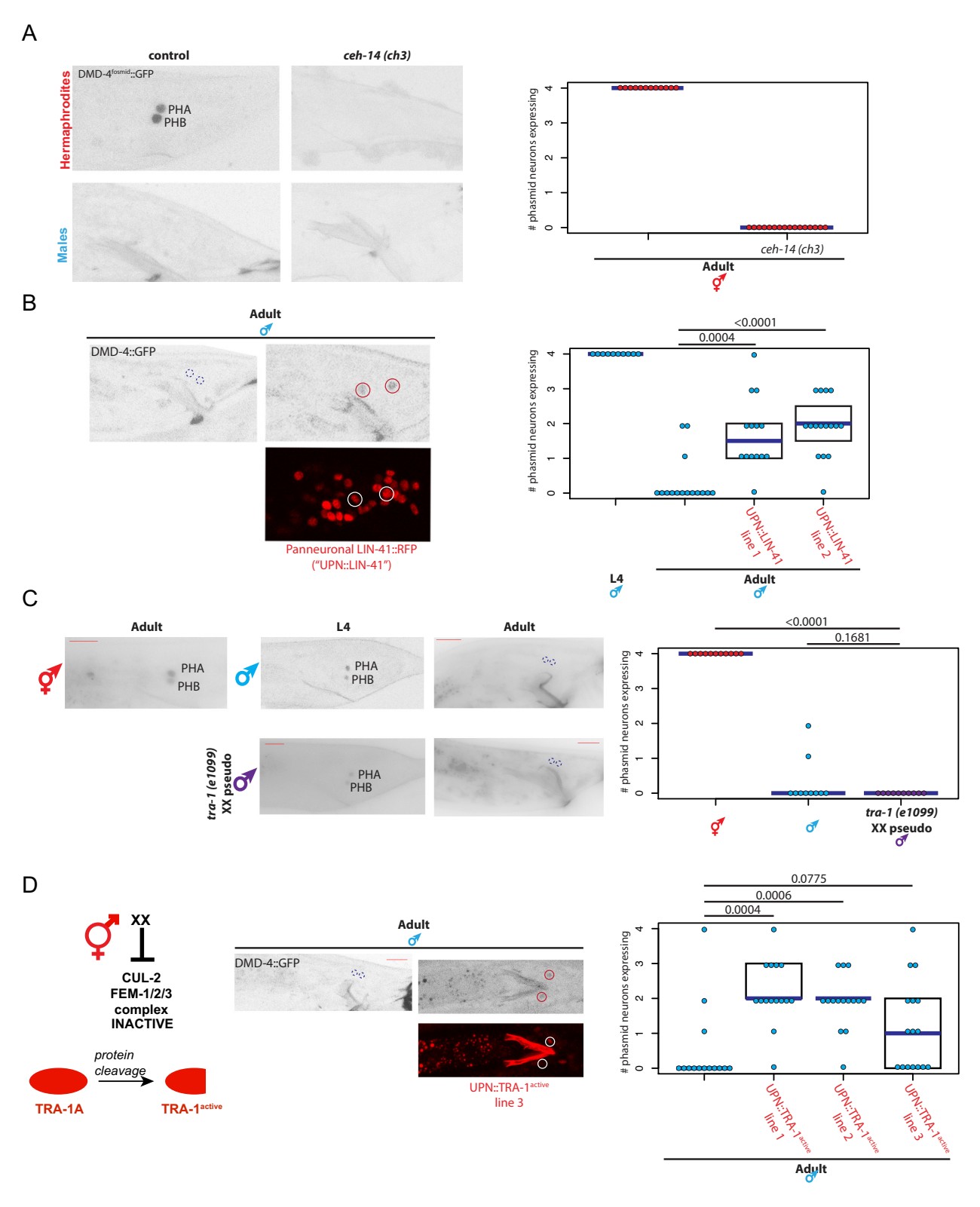

**Figure 5.** *dmd-4* is regulated by spatial, temporal, and sex-specific cues in the phasmid neurons. (**A**) DMD-4 expression is lost in hermaphrodite animals when the *ceh-14* terminal selector of phasmid neuron fate is mutated. Male expression is unaffected. Color-inverted black and white GFP images are shown to the left in all panels, quantification of phasmid neurons expressing DMD-4 in wild-type and mutant animals is shown to the right in all panels. Horizontal blue line indicates median. (**B**) Constitutive, panneuronal expression of the LIN-41 heterochronic gene maintains DMD-4 in the

*Figure 5 continued on next page*

Figure 5 continued

adult male phasmids. Color-inverted black and white GFP images of adult males are shown in addition to the panneuronal, RFP-tagged LIN-41 overexpression ('UPN::LIN-41'). L4 male quantification of DMD-4 expression in the phasmids (from *Figure 1B*) is shown again for reference comparison to control adults (siblings of transgenic animals) and LIN-41 overexpressing males. We quantified two independent transgenic lines. *P* values are shown above horizontal black bars by Wilcoxon rank-sum test with Bonferroni corrections for multiple testing where applicable (see Materials and methods) in all panels. (C) DMD-4 is degraded in the phasmid neurons of *tra-1 (e1099)* XX pseudomales in adulthood, but not in the L4 stage. (D) Constitutive expression of proteolytically cleaved TRA-1$^{active}$ feminizes adult male phasmid neurons and maintains DMD-4 expression. The hermaphrodite sex determination pathway represses the CUL-2/FEM-1/2/3 ubiquitin ligase pathway, resulting in TRA-1 stabilization and proteolytic cleavage, which is schematized to the left. Color-inverted black and white GFP images of adult males are shown in addition to the panneuronal, RFP-tagged TRA-1$^{active}$ in the center ('UPN::TRA-1$^{active}$::RFP'). We examined three independent transgenic lines; two had a significant effect on DMD-4 expression in the male phasmid neurons and one did not.

indeed be made responsible for DMD-4::GFP downregulation and reveal an unanticipated regulatory mechanism.

## The ubiquitin-binding DMA domain controls DMD-4 stability

To assess which part of the protein may be responsible for sexually dimorphic DMD-4 protein expression, we turned to the phylogenetically conserved DMA domain (for **DM a**ssociated). The DMA domain is found associated with the DM domain in most of the phylogenetically conserved DMRT proteins and is also present in the sole homolog of the most primitive DMRT protein from corals (*Miller et al., 2003*). However, the function of the DMA domain has not previously been examined. Pairwise alignments of profile hidden Markov models using HHsearch (*Soding, 2005*) reveals that the DMA domain shows sequence similarity to CUE domains (for **c**oupling of **u**biquitin to **E**R degradation), a subtype of the **ub**iquitin-**a**ssociated (UBA) family of domains (*Kang et al., 2003*; *Figure 7A*). We produced a homology model of the DMD-4 DMA domain based on known CUE-ubiquitin domain structures (*Kang et al., 2003*) using the program I-TASSER (*Zheng et al., 2019*). The homology model showed an appropriate domain structure, with all inwardly oriented residues of hydrophobic character arranged as a well-packed core, with charged and polar residues at the surface (*Figure 7B*). The DMA domain contains three alpha-helices; by analogy with CUE domains, helices 1 and 3 would be expected to form a binding surface for direct non-covalent contact with ubiquitin.

To assess the ability of the DMD-4 DMA domain to bind ubiquitin, we expressed the DMA domain as a maltose-binding protein (MBP) fusion in bacteria, and assessed its binding to human ubiquitin (which differs in only one of the 76 amino acids from *C. elegans* ubiquitin) by Surface Plasmon resonance (SPR) analysis. The DMD-4 DMA domain bound monoubiquitin with an affinity of ~400 µM, which is similar to known affinities for CUE-ubiquitin binding (*Hurley et al., 2006*; *Figure 7C*). CUE domains have also been found to dimerize (*Azurmendi et al., 2010*; *Prag et al., 2003*). Testing whether the DMA domain of DMD-4 may display the same feature, we used size exclusion chromatography with multi-angle static light scattering (SEC-MALS). We found that the DMD-4 DMA domain behaves as a monomer (*Figure 7—figure supplement 1*).

To test the functional significance of ubiquitin binding, we used our homology model to identify Leu$^{31}$ (numbering relative to the beginning of the DMA domain) as a residue expected to make direct contact with ubiquitin (*Figure 7B*) and mutated this residue to Arg in order to disrupt ubiquitin binding. Based on the outward-facing position of this residue in our homology model, we do not anticipate that mutating it results in structural changes to the domain. As expected, the in vitro binding of the mutated DMA domain to ubiquitin was disrupted, with the affinity of mutant DMD-4 DMA for ubiquitin reduced from 404 µM to >3 mM (*Figure 7C*). We engineered this mutation into the endogenous *gfp*-tagged *dmd-4* locus (*ot935*) using CRISPR/Cas9 to generate the *dmd-4 (ot990ot935)* allele. Our guiding hypothesis was that through an abrogation of ubiquitin binding, we may perhaps prevent the degradation of DMD-4 in male PHA/PHB neurons. Much to our surprise, we observed the exact opposite result; we find that in *dmd-4(ot990ot935)* animals, DMD-4 expression is lost in the phasmid neurons in both sexes and at all stages of development (*Figure 7D*). Expression in the head sensory neurons was also lost, again in both sexes of all stages. Consistent with the phenotype of the *ot957* null allele, the *ot990* allele also results in defective pruning on the PHB>AVG synaptic connection in hermaphrodites (*Figure 7—figure supplement 1A*). This suggests

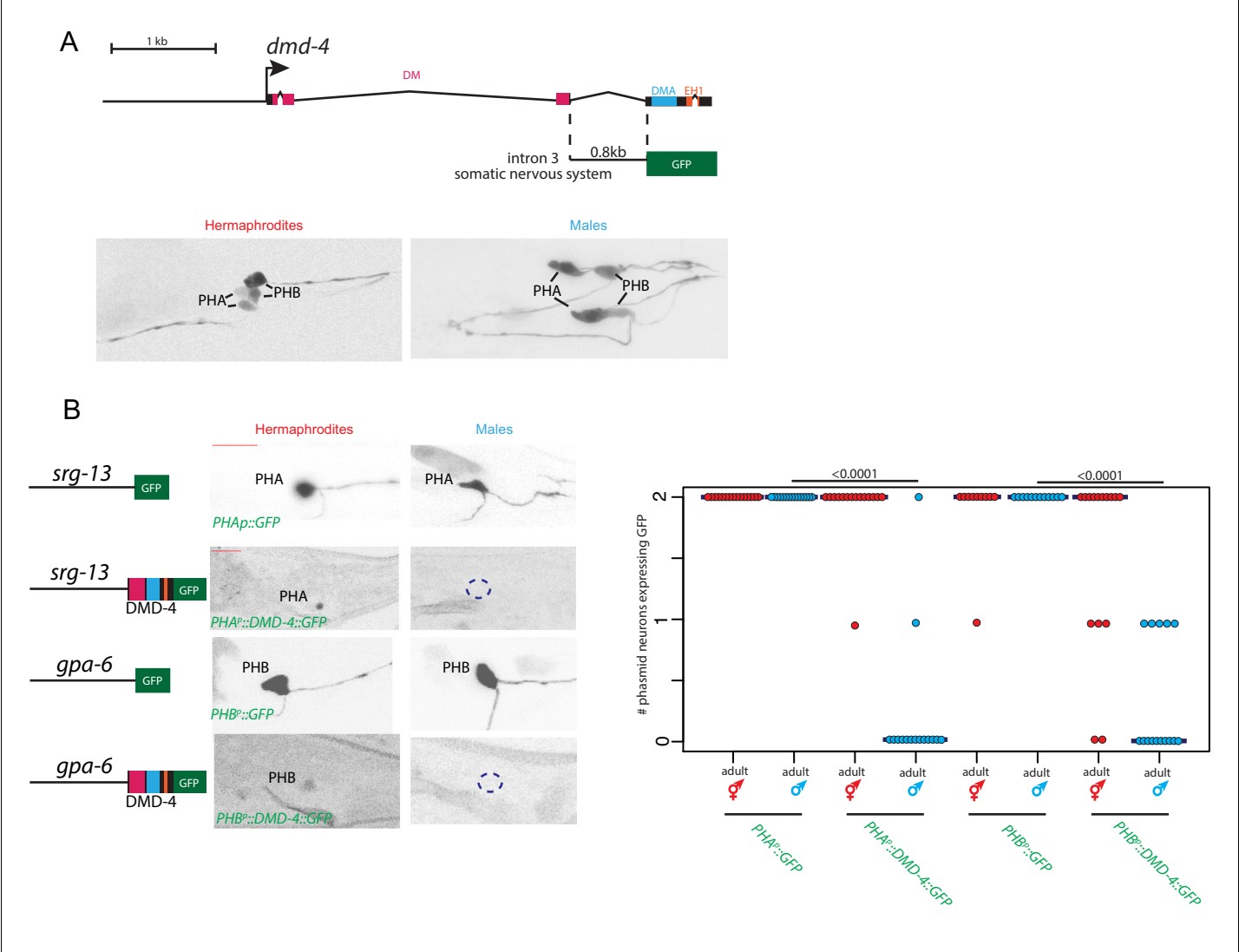

**Figure 6.** Sexually dimorphic degradation of DMD-4. (**A**) Dimorphic expression is not regulated on the level of transcription. The enhancer driving DMD-4 expression in the somatic nervous system is schematized above (same as shown in *Figure 2B*), color-inverted black and white GFP images in adult hermaphrodite and male are shown below. (**B**) Driving *dmd-4* cDNA under heterologous promoters in the phasmid neurons recapitulates the endogenous expression pattern. Heterologous promoters driving *gfp* with and without the *dmd-4* cDNA are shown to the left in adult hermaphrodites and males, quantification is shown to the right. Scale bar (horizontal red bar) denotes 10 microns. For quantification, each dot represents the number of phasmid neurons expressing GFP in one animal (maximum is two neurons for PHAL/R or PHBL/R), red = hermaphrodite and cyan = male. Horizontal blue bars indicate median. *P* values are shown above horizontal black bars by Wilcoxon rank-sum test.

that the binding of the DMD-4 DMA domain to ubiquitin is required to promote the stability of DMD-4 in the nervous system, and that degradation of DMD-4 in the male phasmids at the larval to adult transition is possibly generated by loss of the ubiquitin-based stabilization mechanism.

Intriguingly, the DMA domain mutation that disrupts ubiquitin binding does not affect expression or DMD-4::GFP stability in the alimentary system (pharynx and hmc) throughout embryonic and larval development; however, expression is reduced at the larval to adult transition (*Figure 7D*). Hence, these animals do not display the embryonic lethality associated with *dmd-4* null alleles and in spite of the reduction of protein in the adult, they show no stuffed anterior intestine (*Figure 2—figure supplement 2A, B*).

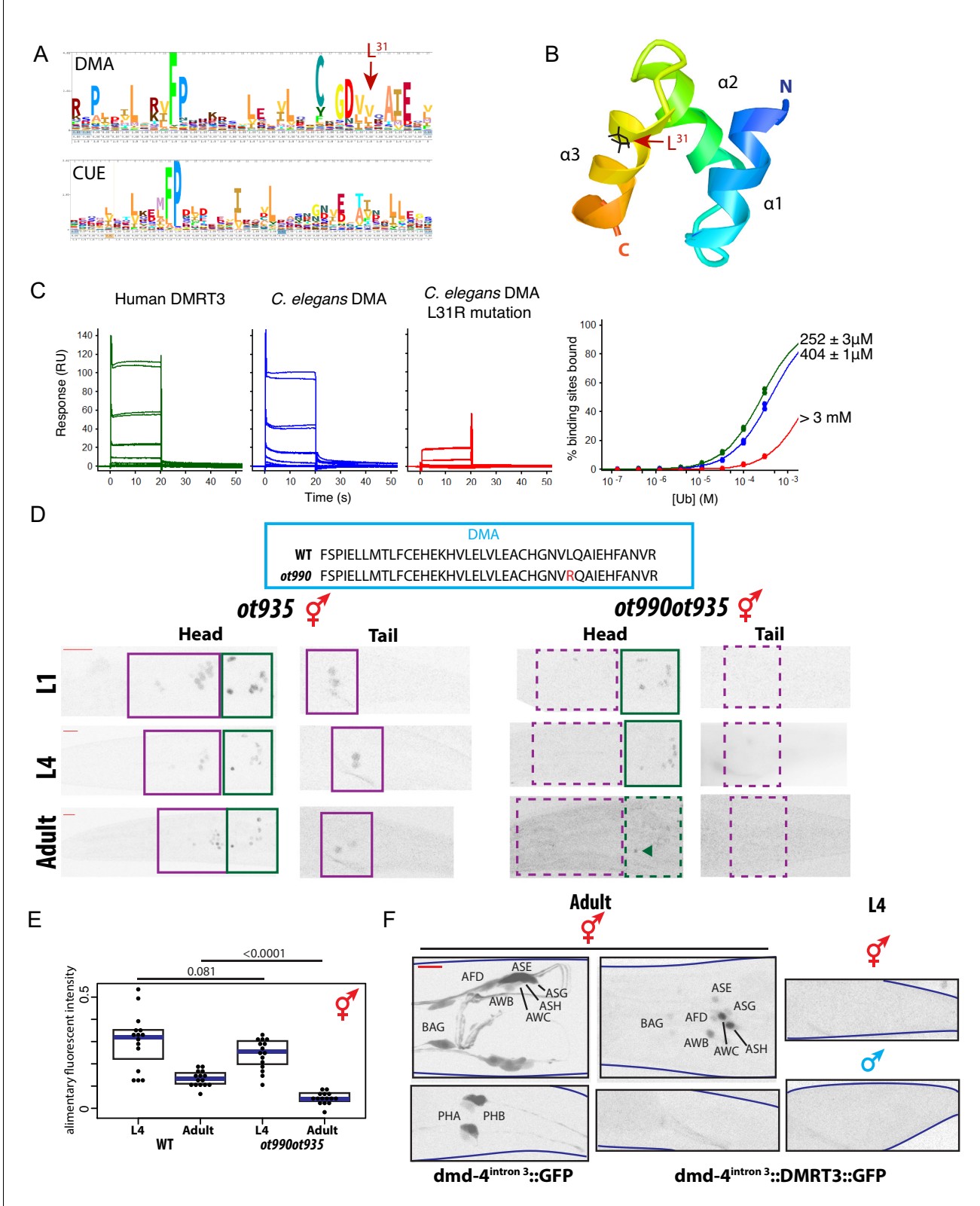

**Figure 7.** The conserved DMA domain is a novel, ubiquitin binding degron. (**A**) Weighted sequence alignment of DMA domains and CUE domains across phyla. The similarities behind these two sequence profiles is demonstrated by pairwise alignments of profile hidden Markov models using HHsearch (*Soding, 2005*) and is noted in the PF03474 Pfam record of the DMA domain. (**B**) Homology model of the DMD-4 DMA domain based on the known structure of a ubiquitin-binding CUE domain (*Kang et al., 2003*). The three alpha-helices are labeled; helices 1 and 3 form a binding surface for

*Figure 7 continued on next page*

*Figure 7 continued*

contact with ubiquitin. The leucine 31 side chain mutated in panel **D** is highlighted in black; note the position on the ubiquitin-binding surface. (**C**) SPR binding analysis of the *C. elegans dmd-4* and human Dmrt3 DMA domains shows that both bind to monoubiquitin, and this binding is lost in the L31R mutation (see panel **A**). (**D**) CRISPR/Cas9 mutation of leucine 31 (see panel **B**) to arginine, which we predicted to impair DMA binding to ubiquitin, promotes DMD-4::GFP degradation. *ot935* is the *gfp*-tagged *dmd-4* locus and the *ot990* allele carries the leucine 31 mutation (to Arg) as indicated. Color-inverted black and white GFP images of L1/L4/adult hermaphrodite heads and tails are shown below, purple boxes indicate nervous system expression, green boxes indicate pharyngeal expression. Dashed boxes indicate missing expression. I5 sometimes maintains DMD-4::GFP expression in adults (green arrowhead). Scale bars (red horizontal bars) denote 10 microns. (**E**) DMD-4::GFP fluorescence intensity in *gfp* tagged animals (*ot935*) and *ot935* animals in which the *ot990* mutation is incorporated (shown in panel **D**) is quantified in a defined region of the posterior pharynx (alimentary tissues excluding I5) in L4 and adult wildtype and mutant hermaphrodites. Each dot represents fluorescence intensity in one animal (normalized to background fluorescence), blue bars indicate medians, black boxes indicate quartiles. *P* values are shown above horizontal black bars by Wilcoxon rank-sum test. (**F**) Expressing human DMRT3 under the *dmd-4*^*intron3*^ enhancer ("*dmd-4*^*intron3*^::*DMRT3::GFP*") recapitulates DMD-4 expression in the head neurons, but in the phasmid neurons DMRT3 is degraded constitutively and is absent in both sexes in both larval and adult stages (compared to expression of the enhancer alone, '*dmd-4*^*intron3*^::*GFP*'). Color-inverted black and white GFP images of adult hermaphrodite heads and tails and L4 hermaphrodite and male tails are shown, red scale bar indicates 10 microns.

The online version of this article includes the following figure supplement(s) for figure 7:

**Figure supplement 1.** Analysis of DMA domain dimerization.
**Figure supplement 2.** Disrupting the DMA domain promotes DMD-4::GFP degradation.

We sought to recapitulate these results independently with other mutations in the DMA domain. Rather than introducing a mutation that is predicted to leave the overall structure of the DMA domain intact, we used CRISPR/Cas9 to mutate the first alpha helix of the DMA domain, which is predicted to affect the overall structural integrity of the DMA domain fold. We isolated three different small insertions/deletions in this helix, *ot966*, *ot967* and *ot968*, each in the context of the *gfp*-tagged *dmd-4* allele (**Figure 7—figure supplement 1B**). We find that these alleles behave similarly to the Leu>Arg missense mutation that is predicted to disrupt ubiquitin binding. DMD-4::GFP is destabilized in the somatic nervous system of both sexes at all stages, but expression in alimentary tissue is unaffected during embryonic and larval development (**Figure 7—figure supplement 1B**).

Taken together, the stability of DMD-4 in both neurons and non-neuronal tissues is promoted by ubiquitin binding to the DMA domain, but the need to promote DMD-4 stability by ubiquitin binding is different in different tissues and at different times (permanently in the nervous system, but not until adulthood in the alimentary system).

### Phylogenetic conservation of DMA domain properties

To test whether the DMA domain of vertebrate DMRT proteins may carry properties similar to the DMD-4 DMA domain, we examined DMRT3, the closest vertebrate homolog of DMD-4 (**Wexler et al., 2014**). We find that the purified DMA domain of human DMRT3 also binds ubiquitin in vitro through SPR analysis, with an affinity similar to that of the DMD-4 DMA domain (**Figure 7E**). However, unlike DMD-4, but like other CUE domains, the DMRT3 domain can homodimerize (**Figure 7—figure supplement 1**).

To test whether the full-length human DMRT3 (hDMRT3) protein is subjected to a similar post-transcriptional regulation as *C. elegans* DMD-4, we used the neuronal enhancer of *dmd-4* to express GFP-tagged hDMRT3 in the *dmd-4(+)* neurons. We found that like DMD-4, hDMRT3::GFP is not degraded in the amphid head neurons, but that in phasmid neurons hDMRT3::GFP is degraded (**Figure 7F**). Unexpectedly, this degradation appears to be not subjected to sexual or temporal control, as we fail to observe hDMRT-3::GFP expression in the phasmid neurons of both sexes at all times (**Figure 7F**).

### Discussion

Sexual differentiation is controlled by an exceptionally diverse range of genetic mechanisms within the animal kingdom. Members of the DMRT transcription factor family are presently the only known factors commonly involved in sexual differentiation across the animal kingdom (**Kopp, 2012**; **Matson and Zarkower, 2012**). However, the DMRT family has radiated in distinct animal phyla,

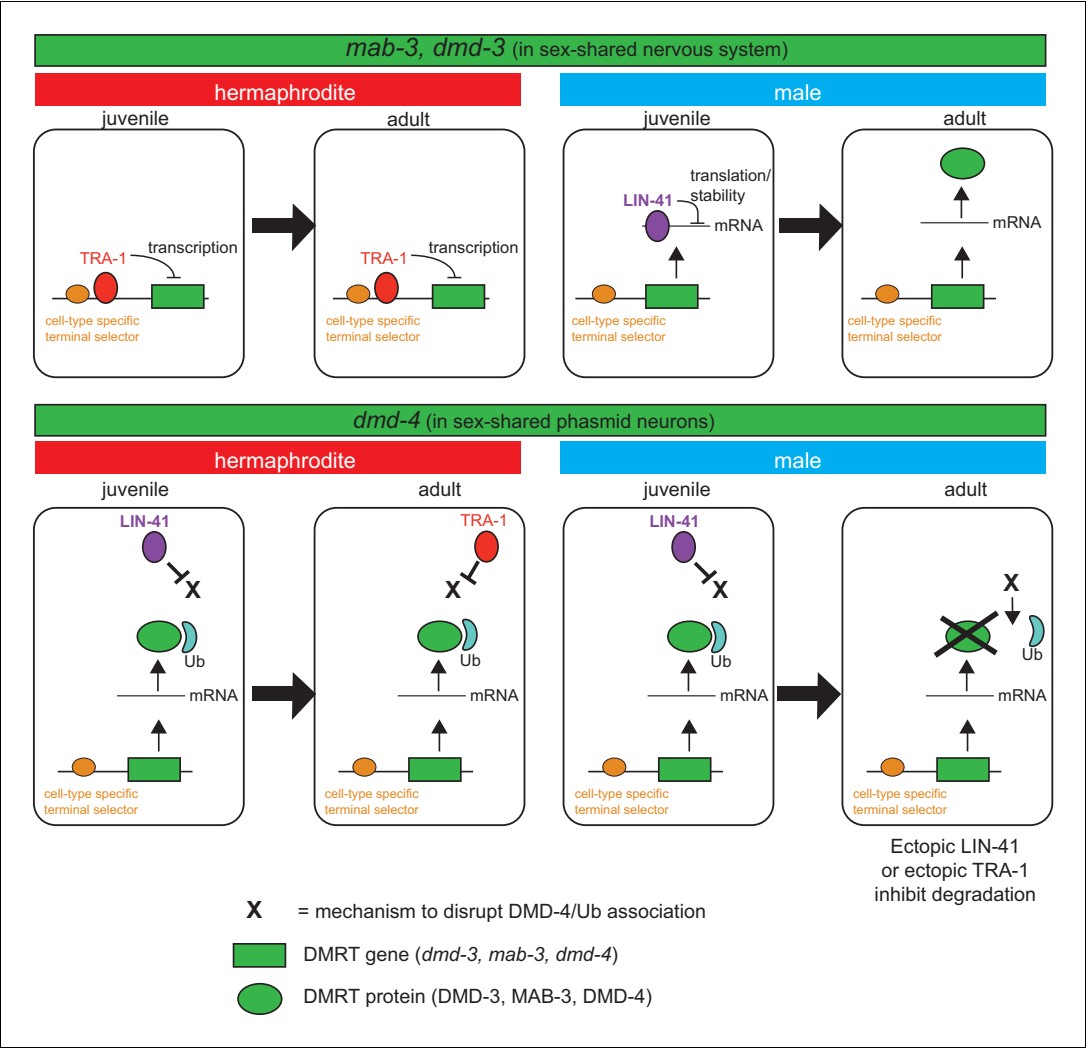

**Figure 8.** Spatiotemporal specificity and regulation of DMD-4 expression differs from the regulation of other *C. elegans* DMRT genes. Similar pathways are used to control the spatial, temporal and sexual specificity of *C. elegans* DMRT genes, but these pathways are utilized in a fundamentally distinct manner for different DMRT proteins, ranging from transcriptional to posttranslation regulation. See text for more discussion.

generating clade-specific family members. For example, the well-studied *doublesex* gene is an arthropod-specific DMRT gene (*Mawaribuchi et al., 2019*). We have described here the expression and function of a DMRT gene, *C. elegans dmd-4,* that in contrast to *doublesex* is conserved throughout the animal kingdom and find that this gene plays a critical role in sculpting sexually dimorphic synaptic connectivity patterns. DMD-4 protein is expressed and functions sex-specifically in the phasmid sensory neurons where it is required to promote hermaphrodite-specific synapse pruning.

We have previously reported that other DMRT proteins (*dmd-5* and *dmd-11*) also control sexually dimorphic synaptic pruning in the phasmid circuit (*Oren-Suissa et al., 2016*), but the function and expression of DMD-4 differs from the previously described cases in three ways. First, loss of DMD-4 results in aberrant maintenance of a normally sexually dimorphic synaptic connection, whereas loss of DMD-5 and DMD-11 resulted in ectopic pruning of a synaptic connection. Second, DMD-5 and DMD-11 function in a post-synaptic neuron (the AVG interneuron, a sex-specific target of the phasmid neurons) to modulate synaptic pruning, whereas we find that DMD-4 functions pre-synaptically. Finally, DMD-4 is present specifically in hermaphrodites, which stands in contrast not only to the male-specificity of DMD-5 and DMD-11 and all other dimorphically expressed DMRT proteins in *C.*

*elegans* (*mab-3*, *mab-23*, *dmd-3*)(*Lints and Emmons, 2002*; *Mason et al., 2008*; *Oren-Suissa et al., 2016*; *Yi et al., 2000*).

In addition to the sexually dimorphic functions in the hermaphrodite phasmid neurons, we found that *dmd-4* is required for animal viability, apparently due to a non-sexually dimorphic requirement in the feeding organ of the worm. The *dmd-4* orthologs DMRT93B (Arthropods) and Dmrt3 (vertebrates) have also been implicated in development of feeding organs based on both expression and function, with mouse Dmrt3 knockout mutants unable to survive past weaning due to feeding defects (*Ahituv et al., 2007*; *Inui et al., 2017*; *Panara et al., 2019*). Apart from non-dimorphic expression in the alimentary system, DMD-4 is also expressed in head sensory neurons of both sexes. While we did not observe a requirement of DMD-4 for general features of neuron specification, we have not examined all possible functions of DMD-4, for example, the synaptic wiring of these cells.

Perhaps the most unanticipated results of our studies relate to the control of sexual specificity of DMD-4 function in phasmid neurons via sex-specific protein degradation of DMD-4. Sex-specific transcription factor degradation is not unprecedented. Not only has it been observed for the TRA-1 master regulator of sexual identity in *C. elegans* (*Schvarzstein and Spence, 2006*), but a recent study in *Drosophila* also revealed sex-specific degradation of the Lola transcription factor (*Sato et al., 2019*). We reveal here a different mechanism that involves an evolutionarily ancient domain, the DMA domain. Phylogenetic analysis of the DMRT gene family suggests that the presence of a DMA domain was characteristic of the three ancestral gene clusters, and arose before the evolution of eumetazoans, but this domain has not been maintained in all extant DMRT clusters (*Mawaribuchi et al., 2019*; *Wexler et al., 2014*). In *C. elegans*, *dmd-4* represents the only gene that encodes a DMA domain, which we identify here as required for protein stability and able to bind monoubiquitin, consistent with its sequence similarity to the CUE domains of the UBA family. We find that the *H. sapiens* Dmrt3 DMA domain is also able to non-covalently bind monoubiquitin and that *H. sapiens* Dmrt3 appears to also be subjected to posttranslational regulation in worms, suggesting a conserved molecular function of the DMA domain. CUE domains have been associated with a breadth of regulatory functions as a result of their association with ubiquitin: dimerization, subcellular trafficking, covalent ubiquitylation, and protein complex formation (*Bagola et al., 2013*; *Chen et al., 2006*; *Davies et al., 2003*; *Prag et al., 2003*). Intriguingly, certain types of non-covalent ubiquitin-binding domains have been shown to stabilize proteins (*Flick et al., 2006*; *Heessen et al., 2005*; *Heinen et al., 2011*; *Heinen et al., 2010*). For example, the non-covalent ubiquitin-binding domain of a yeast transcription factor, Met4, interacts with ubiquitin molecules that are covalently attached elsewhere on the protein to prevent polyubiquitin chain extension and proteasomal degradation of Met4 (*Flick et al., 2006*). Consequently, disruption of this non-covalent ubiquitin binding domain of Met4 results in degradation of Met4. Based on the Met4 precedent, we hypothesize that ubiquitin binding to DMD-4 may also prevent protein degradation.

The apparently ubiquitin-dependent stabilization of DMD-4 is susceptible to three parameters: sexual identity, cell-type and time. In wild-type animals, DMD-4 is only degraded in the phasmid neurons, not in other neuronal or non-neuronal cell types. In addition, this degradation only occurs in males and it only occurs at a specific time of development. Based on the degradation of DMD-4 in phasmid neurons in both sexes at all times upon mutation of the presumptive ubiquitin binding site , we hypothesize that the DMD-4/ubiquitin interaction may be subject to dynamic control in the phasmid neurons of wildtype animals. For example, this stabilizing effect may be disrupted sex-specifically in males, thereby permitting degradation of the protein.

Both the heterochronic regulatory pathway as well as the sex determination pathway control the dynamic nature of this process. Since the forced expression of LIN-41 (and hence the forced retention of a juvenile state) suppresses DMD-4 degradation, it appears that an adult-specific process, controlled by the heterochronic pathway (e.g. a protein phosphorylation event) allows the disruption of the DMD-4/ubiquitin interaction. This destabilization may be the 'default state', but may be antagonized by LIN-41 in juveniles in both sexes and then by TRA-1 in adult hermaphrodites (schematized in *Figure 8*). Consistent with this notion, either forced expression of LIN-41 or TRA-1 prevents destabilization in male adult phasmid neuron.

Strikingly, in the head sensory neurons, there appears to be no dynamic control of stability. In that cellular context, DMD-4 may always require ubiquitin binding for its stabilization. However, in the pharynx and hmc, the situation is again different. While ubiquitin binding is also required to

stabilize the protein, this binding is only required in the adult stage to stabilize DMD-4 in both sexes. We infer this from the observation that the ubiquitin binding mutation only destabilizes DMD-4 in adult, but not in juvenile stages. While we clearly do not yet fully understand the mechanisms that lead to this spatiotemporal regulation of DMD-4 protein level, our studies have identified novel means by which a DMRT protein in specific, and sexual dimorphisms more broadly, are controlled in the nervous system in a manner that depends on cell type, time of development and sexual identity. The deeply conserved nature of the DMA domain suggest that these regulatory mechanisms are conserved in other species as well.

It is fascinating to note the diversity of mechanisms by which the spatial, temporal and sexually specificity of DMRT proteins is controlled, particularly in the context of one and the same species. While similar pathways are used to control the spatial, temporal and sexual specificity, some of these regulatory factors are utilized in a fundamentally distinct manner, as schematized in *Figure 8* for different DMRTs in *C. elegans*. As demonstrated before for *mab-3, dmd-3* and here for *dmd-4*, the spatial specificity of expression (i.e. the cell types which display dimorphic DMRT expression) is controlled by terminal selector-type transcription factors that generally control neuronal identity (*Pereira et al., 2019*). The heterochronic pathway and the sex determination pathway then sculpt the sexual and temporal specificity of these genes, but in a mechanistically different manner. The sexual specificity of *mab-3* and *dmd-3* is controlled by direct transcriptional repression by the TRA-1 protein, which prevents transcription of these genes in hermaphrodites (*Mason et al., 2008*; *Pereira et al., 2019*; *Yi et al., 2000*). Temporal specificity is controlled by the posttranscriptional repression of *mab-3* and *dmd-3* mRNA via a member of the heterochronic pathway (specifically, the mRNA binding protein LIN-41), therefore resulting in male- and adult-specific expression of MAB-3 and DMD-3 protein (*Figure 8*; *Pereira et al., 2019*). In striking contrast, *dmd-4* is transcribed and the protein accumulates in both sexes in juvenile states, but its sexual specificity is regulated by protein stabilization and it appears to be this protein stabilization phenomenon that is controlled by presently unknown means via the LIN-41 and TRA-1 protein (*Figure 8*). The largely unexplored patterns of expression of vertebrate DMRT proteins - particularly those that contain a DMA domain (DMRT3, DMRT4, DMRT5) - may reveal similarly complex pattern of regulatory control.

# Materials and methods

## Key resources table

| Reagent type (species) or resource | Designation | Source or reference | Identifiers | Additional information |
|---|---|---|---|---|
| Genetic reagent (*Caenorhabditis elegans*) | *him-5 (e1490) V; dmd-4 (ot935) X* | This work | OH15814 'DMD-4::GFP' | *Figure 1* |
| Genetic reagent (*C. elegans*) | *dmd-4 (ot896) X* | This work | OH15302 | *Figure 2* |
| Genetic reagent (*C. elegans*) | *dmd-4(ot933)/tmC24 X* | This work | OH16260 | *Figure 2* |
| Genetic reagent (*C. elegans*) | *ccIs4251; stIs10539* | CGC | SD1633 | *Figure 2* *ccIs4251 [(pSAK2) myo-3p::GFP::LacZ::NLS + (pSAK4) myo-3p::mitochondrial GFP + dpy-20(+)] I; stIs10539 [dmd-4p::HIS-24::mCherry + unc-119(+)]* |
| Genetic reagent (*C. elegans*) | *otEx6774; him-5 (e1490)* | This work | OH14506 | *Figure 2* dmd-4int3::GFP, pRF4 |
| Genetic reagent (*C. elegans*) | *him-5 (e1490) V; dmd-4 (ot957ot935) X* | This work | OH15908 | *Figure 2* |

*Continued on next page*

*Continued*

| Reagent type (species) or resource | Designation | Source or reference | Identifiers | Additional information |
|---|---|---|---|---|
| Genetic reagent (*C. elegans*) | *otIs612; him-5 (e1490)* | *Oren-Suissa et al., 2016* | OH13575 'PHB > AVA' | *Figure 3* *MVC12 (flp-18p::nlg-1::gfp11) 15 ng/ul, MVC6 (gpa-6p::nlg-1::gfp1- 10) 15 ng/ul, MVC11 (flp- 18 p::mcherry) 10 ng/ul, MVC15 (nlp- 1::mcherry) 10 ng/ul, pRF4 50 ng/ul* |
| Genetic reagent (*C. elegans*) | *otIs612; him-5 (e1490); dmd-4 (ot957ot935) X* | This work | OH16014 'PHB > AVA' | *Figure 3* *MVC12 (flp-18p:: nlg-1::gfp11) 15 ng/ul, MVC6 (gpa-6p::nlg-1::gfp1- 10) 15 ng/ul, MVC11 (flp- 18 p::mcherry) 10 ng/ul, MVC15 (nlp- 1::mcherry) 10 ng/ul, pRF4 50 ng/ul* |
| Genetic reagent (*C. elegans*) | *otEx6347; him-5 (e1490) V* | *Oren-Suissa et al., 2016* | OH13696 'PHA > AVG' | *Figure 3* *srg-13::BirA::nrx-1 25 ng/ul, inx-18p::AP::nlg-1 25 ng/ul, inx-18p::wcherry 10 ng/ul, unc-122::streptavidin:: 2xsfGFP 25 ng/ul, pRF4 50 ng/ul* |
| Genetic reagent (*C. elegans*) | *otEx6347; him-5 (e1490) V; dmd-4 (ot957ot935) X* | This work | OH16067 'PHA > AVG' | *Figure 3* *srg-13::BirA::nrx-1 25 ng/ul, inx-18p::AP::nlg-1 25 ng/ul, inx-18p::wcherry 10 ng/ul, unc-122::streptavidin:: 2xsfGFP 25 ng/ul, pRF4 50 ng/ul* |
| Genetic reagent (*C. elegans*) | *otIs614; him-5 (e1490)* | *Oren-Suissa et al., 2016* | OH13577 'PHB > AVG' | *inx-18p::NLG-1::GFP11 30 ng/ul, MVC6 gpa-6::NLG-1:::GFP1-10 30 ng/ul, MVC15 nlp-1::mcherry 5 ng/ul, inx-18::wcherry 10 ng/ul, pRF4 50 ng/ul* |
| Genetic reagent (*C. elegans*) | *otIs614; him-5 (e1490) V; dmd-4 (ot957ot935) X* | This work | OH16017 'PHB > AVG' | *Figure 3* *inx-18p::NLG-1::GFP11 30 ng/ul, MVC6 gpa-6:: NLG-1:::GFP1-10 30 ng/ul, MVC15 nlp-1::mcherry 5 ng/ul, inx-18::wcherry 10 ng/ul, pRF4 50 ng/ul* |
| Genetic reagent (*C. elegans*) | *otEx7431; otIs614; him-5 (e1490) V; dmd-4 (ot957ot935) X* | This work | OH16217 'PHBᵖ::DMD-4 line 1' | *Figure 3* *gpa-6p::DMD-4::GFP, ttx-3p::GFP* |
| Genetic reagent (*C. elegans*) | *otEx7433; otIs614; him-5 (e1490) V; dmd-4 (ot957ot935) X* | This work | OH16208 'PHBᵖ::DMD-4 line 2' | *Figure 3* *gpa-6p::DMD-4::GFP, ttx-3p::GFP* |
| Genetic reagent (*C. elegans*) | *otIs702; him-5 (e1490) V* | This work | OH15565 'PHA::RAB-3' | *Figure 3* *srg-13p::RAB-3::BFP, srg-13p::gfp, ttx-3::mCherry* |
| Genetic reagent (*C. elegans*) | *him-5 (e1490) V; dmd-4 (ot935ot957) X* | This work | OH16030 'PHA::RAB-3' | *Figure 3* *srg-13p::RAB-3::BFP, srg-13p::gfp, ttx-3::mCherry* |
| Genetic reagent (*C. elegans*) | *tax-4 (p678); him-5 (e1490)* | *Oren-Suissa et al., 2016* | OH15339 | *Figure 4* |
| Genetic reagent (*C. elegans*) | *tax-4 (p678); him-5 (e1490) V; dmd-4 (ot957ot935) X* | This work | OH16115 | *Figure 4* |
| Genetic reagent (*C. elegans*) | *wgIs418; him-5 (e1490) V* | This work | OH16268 | *Figure 5* DMD-4$^{fosmid}$::gfp |

*Continued on next page*

*Continued*

| Reagent type (species) or resource | Designation | Source or reference | Identifiers | Additional information |
|---|---|---|---|---|
| Genetic reagent (C. elegans) | wgIs418; ceh-14 (ch3) X | This work | OH14503 | *Figure 5* DMD-4$^{fosmid}$::gfp |
| Genetic reagent (C. elegans) | otEx7429; him-5 (e1490) V; dmd-4 (ot935) X | This work | OH16159 'UPN::LIN-41 line 1' | *Figure 5* UPN::LIN-41A, rab-3p:: tagRFP, pRF4 |
| Genetic reagent (C. elegans) | otEx7451; him-5 (e1490) V; dmd-4 (ot935) X | This work | OH16235 'UPN::LIN-41 line 2' | *Figure 5* UPN::LIN-41A, rab-3p:: tagRFP, pRF4 |
| Genetic reagent (C. elegans) | tra-1 (e1099) III/hT2 [bli-4(e937) umnIs36] (I;III); dmd-4 (ot935) X | This work | OH16194 | *Figure 5* |
| Genetic reagent (C. elegans) | otEx7352; him-5 (e1490) V; dmd-4 (ot935) X | This work | OH15952 'UPN::TRA-1$^{active}$ line 1' | *Figure 5* UPN::tagRFP:: TRA-1$^{active}$, pRF4 |
| Genetic reagent (C. elegans) | otEx7353; him-5 (e1490) V; dmd-4 (ot935) X | This work | OH16158 'UPN::TRA-1$^{active}$ line 2' | *Figure 5* UPN::tagRFP:: TRA-1$^{active}$, pRF4 |
| Genetic reagent (C. elegans) | otEx7354; him-5 (e1490) V; dmd-4 (ot935) X | This work | OH15993 'UPN::TRA-1$^{active}$ line 3' | *Figure 5* UPN::tagRFP:: TRA-1$^{active}$, pRF4 |
| Genetic reagent (C. elegans) | otIs627; him-8 (e1489) | This work | OH14022 'PHAp::GFP' | *Figure 6* srg-13p::GFP |
| Genetic reagent (C. elegans) | otEx7435; him-5 (e1490) V ; dmd-4 (ot957ot935) X | This work | OH16177 'PHAp::DMD-4::GFP' | *Figure 6* srg-13p::DMD-4:: GFP, ttx-3p::GFP |
| Genetic reagent (C. elegans) | otEx6456; him-5 (e1490) V | This work | OH13892 'PHBp::GFP' | *Figure 6* gpa-6p::GFP |
| Genetic reagent (C. elegans) | otEx7433; him-5 (e1490) V; dmd-4 (ot957ot935) X | This work | OH16175 'PHBp::DMD-4::GFP' | *Figure 6* gpa-6p::DMD-4:: GFP, ttx-3p::GFP |
| Genetic reagent (C. elegans) | him-5 (e1490) V; dmd-4 (ot990ot935) X | This work | OH16125 | *Figure 7* |
| Genetic reagent (C. elegans) | nIs175 [ceh-28p:: 4xNLS::GFP + lin-15AB(+)] IV; ceh-34(n4796) V | CGC | MT15695 | *Figure 2—figure supplement 1* |
| Genetic reagent (C. elegans) | otIs518; him-5 (e1490) V; dmd-4 (ot957ot935) X | This work | OH16054 | *Figure 2—figure supplement 2* |
| Genetic reagent (C. elegans) | ntIs1; dmd-4 (ot957ot935) X | This work | OH16225 | *Figure 2—figure supplement 2* |
| Genetic reagent (C. elegans) | wgIs73; him-5 (e1490) V; dmd-4 (ot957ot935) X | This work | OH16018 | *Figure 2—figure supplement 2* |
| Genetic reagent (C. elegans) | otIs612; unc-13(e51) I; him-5 (e1490) V | This work | MOS14 | *Figure 3—figure supplement 1* |
| Genetic reagent (C. elegans) | otEx6913; unc-13(e51) I; him-5 (e1490) V | This work | MOS234 | *Figure 3—figure supplement 1* inx-18p::NLG-1::GFP11, MVC6 gpa-6::NLG-1::: GFP1-10, MVC15 inx-18:: wcherry, pRF4 |
| Genetic reagent (C. elegans) | etyEx2; otIs614; him-5 (e1490) V | This work | MOS243 | *Figure 3—figure supplement 1* gpa-6::HisCl1, myo-2::GFP |
| Genetic reagent (C. elegans) | him-5 (e1490) V; dmd-4 (ot966ot0935) X | This work | OH16013 | *Figure 7—figure supplement 2* |

*Continued on next page*

*Continued*

| Reagent type (species) or resource | Designation | Source or reference | Identifiers | Additional information |
|---|---|---|---|---|
| Genetic reagent (*C. elegans*) | *him-5 (e1490) V; dmd-4 (ot967ot935) X* | This work | OH16015 | *Figure 7—figure supplement 2* |
| Genetic reagent (*C. elegans*) | *him-5 (e1490) V; dmd-4 (ot968ot935) X* | This work | OH16016 | *Figure 7—figure supplement 2* |
| Genetic reagent (*C. elegans*) | *otIs614; him-5 (e1490) V; dmd-4 (ot990ot935) X* | This work | OH16771 | *Figure 7—figure supplement 2* |
| Genetic reagent (*C. elegans*) | *ccIs4443 IV; him-5 (e1490) V* | This work | OH16545 | *Figure 2—figure supplement 1* arg-1::GFP + dpy-20(+) |
| Genetic reagent (*C. elegans*) | *ccIs4443 IV; him-5 (e1490) V; dmd-4 (ot933) X* | This work | OH16770 | *Figure 2—figure supplement 1* |

## Strains

Wild-type strains were *C. elegans* variety Bristol, strain N2. Worms were maintained by standard methods (*Brenner, 1974*). Worms were grown at 20C on nematode growth media (NGM) plates seeded with bacteria (*E. coli* OP50) as a food source, with the exception of temperature sensitive alleles, which were maintained at 15C. Mutant and transgenic strains used are listed in the Key Resources Table.

## CRISPR/Cas9-mediated genome engineering

The *ot935, ot933,* and *ot990* alleles were generated using Cas9 protein, tracrRNA, and crRNA from IDT, as previously described (*Dokshin et al., 2018*). The GFP repair template for *ot935* was amplified from pPD95.75, and it was fused in-frame immediately before the endogenous *dmd-4* stop codon using gRNA GTTTTCAACCGTATTACTTG.

*ot933* is a 556 bp insertion/deletion beginning at +four relative to the *dmd-4* start codon and was generated using gRNA: TTGTTGTGTATATATCACAT. With 100 bp flanks (underlined), the wild-type sequence is (caps-lock indicates exons):
atctacagaccgagcaagcgaaagaggcggaatcacaaattgttgtgtatatcacattggctattcatttttgttattagctg atctatagataATGATGATCGGTAATCTACATGTATTCCCAAACGGACGAATCGAAAGAG AACGGAAACCTAAATGTGCCAGgtaagagaagctttgtaaactaaatactctaacaacttgcaaccattttgt aaatgttccagATGCAGAAATCACGGACTGGTCAGTTGGCTGAAAGGACACAAGCGGC ATTGCAAGTACAAgtaagttgaagcggaacacttctgcaaaaaatattgtttagtttaagcatttataactagaatatt gtgttacaatggctcagtatcaatcaacattaattcattttgcacacgttctctatgaagaatataggttttggaatggggttaaa atacatgcttcttttttcattgtacataatttatgtttacgattgtgagttataaaaatgtagttttttttctaacaactaatcgacttgata acggttttggtgaaaattgaaaattgaaaaaattcataatttcagtagaatatctaggcacttctaattagtaaaaattgattct gaatgctgaagtagaaaataatttaattttttattaaaataaacgtatacagtatattttttttaaacgattattgataatagtctagtc tcccacattatatatatatatattaatcgttgataaatttgaagtaggataatttataacacaac
And the mutant sequence is (not underlined indicates insertion):
atctacagaccgagcaagcgaaagaggcggaatcacaaattgttgtgtatatcacattggctattcatttttgttattagctg atctatagataATGAaataatcgtttaaaatattttttaaacgattattgataatagtctagtctcccacattatatatatatat taatcgttgataaatttgaagtaggataatttataacacaac

*ot990* is a substitution in the *dmd-4* DMA domain generated using gRNA TGGCTTGAAG TACATTTCCA and a ssODN repair template (*Dokshin et al., 2018*). The mutated DMA nucle-otide sequence is (bold indicates arginine codon):
TTCTCACCAATTGAACTCCTGATGACTTTATTTTGTGAACACGAAAAGCACGTGCTCGAAC TTGTGCTCGAAGCCTGCCATGGAAATGTA**CGT**CAAGCCATTGAACATTTTGCAAATG TGAGA.

The *ot896* and *ot957* deletion alleles were generated using *dpy-5* coCRISPR and Cas9 plasmid as previously described (*Arribere et al., 2014*). *ot896* is a 916 bp deletion/insertion from

+3293 to +4208 relative to the *dmd-4* start codon. gRNAs used were AATCTGCTCGTGGGA TTGAT and TATACACCTACACAGAAAAA.

With 100 bp flanks (underlined), the wild-type sequence is (caps-lock indicates exons):
<u>Ggactatccaatgcccaaaaaatgataaagccagtaacaaaatctttatcattagtgccttgagttcacatgtttcatatatta tttttttattacaagctgaatttgcaaaacgtactctccttctcctttcatttcatcaacttataaattgaataactcgaaaattgatta agtttatcgatatgtttgtgccacttactcgctagttagtgcaacttcgcagttttcatttccggtcagttcatatgcagaaatttac gctcttggtgtttttttttttcaatttgaataattttttgttcaaaaaaaagcaaggattcatctatatctctttatacggagggtggagc aagtaacaggattatcaacttttttttttgcatttatctcgtgagagcttatgtgataagctagcgaaaaaatttcccaagttttttctc tttttttttgaatgaaaatgagtgtttagtgattcccttttctgtgtaggtgtataaatctgctcgtgggattgatcggctaatagaaa aaaatttattttaaactatcgtcctaaaataatatattcag</u>AGTTGTTGCTGGGCAAGCAATTGATAGGCT CCCTCAAGGTCCAGTTTGGAACACAGCTGGTGGTGAAGACGAAGATATGGACTAC TTGGATGAAGAAATCGAAGAAAAACCGCCAACTCCAGTGTCAGAGGAACTCGTCC CTTTGAAACGGAAGAAAGTAGAGGAGACATACGAATTGTCGAGCTTCTCACCAATT GAACTCCTGATGACTTTATTTTGTGAACACGAAAAGCACGTGCTCGAACTTGTGCT CGAAGCCTGCCATGGAAATGTACTTCAAGCCATTGAACATTTTGCAAATGTGAGAC GAGTGAAAAATATTAATCAAATGAAGCTATTCGCTGCAGCAACAAGGATGGATCAC GTGTTTCCAAATATGACAAACTTTATACTTCCAAAGCAATCTTTTCTCATTGATTCGT TGTTAGAG<u>gtaagataaagaaactttaaacttttttataatttatattacatactatctgaaaccggcaattttcag</u>CAAC CAACATTCCCAGTCTCCAGTCAAGCATCTACAAGTACTAAAAC

And the mutant sequence is (no underline indicates insertion):
<u>Ggactatccaatgcccaaaaaatgataaagccagtaacaaaatctttatcattagtgccttgagttcacatgtttcatatatta tttttttattacaagc</u>gagcagatttata<u>tttaaacttttttataatttatattacatactatctgaaaccggcaattttcag</u>CAACC AACATTCCCAGTCTCCAGTCAAGCATCTACAAGTACTAAAAC

*ot957* is a 510 bp deletion removing +3201 to +3710 relative to the *dmd-4* start codon, made in the context of *ot935* using gRNAs TATACACCTACACAGAAAAA and AATTGCTCTTGGAC TACGGT.

With 100 bp flanks (underlined), the wild-type sequence is (caps-lock indicates exons):
<u>atgctagttattgtattacaaattgcacctgtaattctcttgaatcattttttttttcacttgcgatatcaaagtctttatccttcggatgc cggactatc</u>caatgcccaaaaaatgataaagccagtaacaaaatctttatcattagtgccttgagttcacatgtttcatatatt attttttttattacaagctgaatttgcaaaacgtactctccttctcctttcatttcatcaacttataaattgaataactcgaaaattgatt aagtttatcgatatgtttgtgccacttactcgctagttagtgcaacttcgcagttttcatttccggtcagttcatatgcagaaattta cgctcttggtgtttttttttttcaatttgaataattttttgttcaaaaaaaagcaaggattcatctatatctctttatacggagggtggag caagtaacaggattatcaacttttttttttgcatttatctcgtgagagcttatgtgataagctagcgaaaaaatttcccaagttttttct ctttttttttgaatgaaaatgagtgtttagtgattcccttttttctgtgtaggtgtataaatctgctcgtgggattgatc<u>ggctaatagaa aaaaatttattttaaactatcgtcctaaaataatatattcag</u>AGTTGTTGCTGGGCAAGCAATTGATAGGC TCCCTCAAGGTCCAGTTT

And the mutant sequence is (*ot957* is a clean deletion with no inserted sequences):
Atgctagttattgtattacaaattgcacctgtaattctcttgaatcattttttttttcacttgcgatatcaaagtctttatccttcggatgc cggactatcggctaatagaaaaaaatttattttaaactatcgtcctaaaataatatattcagAGTTGTTGCTGGGCA AGCAATTGATAGGCTCCCTCAAGGTCCAGTTT

*ot966*, *ot967*, and *ot968* are small indels in the DMA domain generated using gRNA TGGC TTGAAGTACATTTCCA

The wildtype DMA domain sequence is:
TTCTCACCAATTGAACTCCTGATGACTTTATTTTGTGAACACGAAAAGCACGTGCTC GAACTTGTGCTCGAAGCCTGCCATGGAAATGTACTTCAAGCCATTGAACATTTTGC AAATGTGAGA

The *ot966* sequence is:
TTCTCACCAATT**CGT**G**TTC**AACTCCTGATGACTTTATTTTGTGAACACGAAAAGCAC GTGCTCGAACTTGTGCTCGAAGCCTGCCATGGAAATGTACTTCAAGCCATTGAACA TTTTGCAAATGTGAGA

The *ot967* sequence is:

TTCTCACAACTCCTGATGACTTTATTTTGTGAACACGAAAAGCACGTGCTCGAACTT
GTGCTCGAAGCCTGCCATGGAAATGTACTTCAAGCCATTGAACATTTTGCAAATGT
GAGA

The *ot968* sequence is:
TTCTCACCAATGATGACTTTATTTTGTGAACACGAAAAGCACGTGCTCGAACTTGTG
CTCGAAGCCTGCCATGGAAATGTACTTCAAGCCATTGAACATTTTGCAAATGTGAGA

Resultant amino acid sequences of *ot966*, *ot967*, and *ot968* are given in *Figure 7—figure supplement 2B*.

## Cloning and constructs

The LIN-41A and TRA-1[active] cDNAs used in this study were reverse transcribed from *C. elegans* total RNA using the SuperScript III First-Strand Synthesis System (Thermo-Fisher). TRA-1[active] is a shortened version of the TRA-1A cDNA (860aa) predicted to encode the proteolytically activated version of TRA-1A generated by C-terminal cleavage in hermaphrodites (*Schvarzstein and Spence, 2006*). This truncated protein was previously shown to have feminizing activity when expressed under control of its own regulatory elements (*Schvarzstein and Spence, 2006*).

Restriction-Free Cloning was used to insert TRA-1[active] into UPN::RFP (to generate pEAB70) and LIN-41A into UPN driver vector (to generate pHS41). UPN is a concatenated panneuronal promoter (containing promoter fragments from *unc-11*, *rgef-1*, *ehs-1*, and *ric-19*), and was generated and cloned by Ev Yemini (Yemini et al., submitted). pHS41 [UPN::LIN-41A] was cloned by HaoSheng Sun.

The DMD-4 cDNA was obtained from the Dharmacon *C. elegans* ORF collection and cloned using Restriction-Free Cloning into pPD95.75. Gibson assembly was used to clone the *srg-13* and *gpa-6* promoters (*Oren-Suissa et al., 2016*) into the SphI/XmaI sites.

The human DMRT3 cDNA was codon-optimized (https://worm.mpi-cbg.de/codons/cgi-bin/optimize.py), synthesized by GeneWiz, and cloned using Restriction-Free Cloning into *dmd-4*[intron 3]*::gfp* to generate pEAB71.

## Microscopy

Worms were anesthetized using 100 mM of sodium azide ($NaN_3$) and mounted on 5% agar on glass slides. Worms were analyzed by Nomarski optics and fluorescence microscopy, using a Zeiss 880 confocal laser-scanning microscope. Multidimensional data was reconstructed as maximum intensity projections using Zeiss Zen software. For GRASP and RAB-3 experiments, animals were imaged using 63x objective and puncta were quantified by scanning the original full z-stack for distinct dots in the area where the processes of the two neurons overlap. GRASP and RAB-3 experiments were scored blinded to genotype for mutant analysis and rescue array analysis. For fluorescence quantification experiments, animals were imaged using 40x objective with fixed imaging settings, and quantification was performed on maximum intensity projections using the Zeiss Zen software by quantifying the mean fluorescence intensity in rectangular area of the posterior pharynx (containing all posterior muscle and intestinal valve cells that express DMD-4), and normalizing this value to the image background (an identical rectangular area directly anterior). Sex of L1 animals was determined using rectal epithelial cell morphology.

## Behavioral analysis

### SDS-avoidance behavior

SDS avoidance assay was based on procedures described (*Hilliard et al., 2002*). A small drop of solution containing either the repellent (0.1% SDS in M13 buffer) or buffer (M13 buffer: 30 mM Tris-HCl pH 7.0, 100 mM NaCl, 10 mM KCl) is delivered near the tail of an animal while it moves forward. Once in contact with the tail, the drop surrounds the entire animal by capillary action and reaches the anterior amphid sensory organs. Drop was delivered using 10 uL glass calibrated pipets (VWR international) pulled by hand on a flame to reduce the diameter of the tip. The capillary pipette was mounted in a holder with rubber tubing and operated by mouth. Assayed worms were transferred individually to fresh non-wet unseeded NGM plates. Each assay started with testing the animals with drops of M13 buffer alone. The response to each drop was scored as reversing or not reversing. The

avoidance index is the number of reversal responses divided by the total number of trials. An Inter Stimuli Interval of at least two minutes was used between successive drops to the same animal. Each animal was tested 10 times. Two experimental replicates were performed for each experiment.

### Chemotaxis assays

The response to NaCl gradients was assayed as previously described (*Bargmann and Horvitz, 1991*). Briefly, 10 ml of buffered agar (20 g/l agar, 1 mM $CaCl_2$, 1 mM $MgSO_4$ and 5 mM $KPO_4$) was poured into 10 cm diameter petri dishes. To establish the chemical gradient, we applied 10 μl of odorant to the attractant/repulsion spot and 10 μl of double-distilled $H_2O$ to the control spot. The odorant was allowed to diffuse for 14–16 hr at room temperature before the assay. To increase the steepness of the gradient, another 4 μl of odorant or water was added to the same spots 4 hr before the assay. We applied a 1 μl drop of 1 M sodium azide to both attractant and control spots 10 min before the assay to immobilize worms that reached these areas. Synchronized animals were washed three times with CTX solution (1 mM $CaCl_2$, 1 mM $MgSO_4$ and 5 mM $KPO_4$) and 100–200 animals were placed in the center of the assay plate in a minimal volume of buffer. Animals were allowed to move about the agar surface for 1 hr, after which assay plates were placed at 4°C overnight. The distribution of animals across the plate was then determined and a chemotaxis index was calculated as the number of animals at the odorant minus the number of animals at the control spot, divided by the total number of animals. Animals that did not leave the initial inner circle were not included in the count of total number of animals, as these animals were dead or had movement defects. N2 animals were used as controls.

## Neuronal silencing using the histamine chloride channel system

We silenced the PHB neurons by expressing HisCl1 channel (*Pokala et al., 2014*) in the PHB neurons, using a *gpa-6* driver, as previously described (*Oren-Suissa et al., 2016*). Transgenic animals were picked at the L4 stage and placed on NGM plates containing 10 mM histamine with OP50 bacteria as food source. As a control, animals were placed on NGM plates containing OP50 bacteria but no histamine. GRASP connectivity was assessed ~24 hr later when the animals had reached adulthood.

## Biochemical analysis

### Protein expression and purification

*C. elegans* and *H. sapiens* DMA domain cDNAs were synthesized by GeneWiz, codon optimized for *E. coli*. These cDNAs were cloned into the KpnI site of pMal-c5x (NEB) to produce maltose binding protein fusions, and transformed into BL21 cells. *E. Coli* BL21/DE3 cells were grown in Luria Broth with 100 ug/mL ampicillin. When OD at 600 nm of 1.0 was attained, the cells were induced with 300 uM IPTG and grown for an additional 2 hr. Cells were harvested by centrifugation and sonicated in 20 mM HEPES, pH 7.4, 200 mM NaCl, 1 mM EDTA and protease inhibitors. After clarification by centrifugation, 5 mL of amylose resin (New England BioLabs) equilibrated in HEPES 20 mM pH 7.4, NaCl 200 mM was added to ~60 mL of clear lysate and subjected to gentle rocking at 4°C for 2 hr. The bound resin was packed onto a column and washed with at least 15 column volumes of 20 mM HEPES, pH 7.4, 200 mM NaCl. Proteins were eluted with 30 mL of the HEPES 20 buffer supplemented with 10 mM maltose. This eluent was concentrated and loaded onto a Superose 200 10/300 GL Increase (GE Healthcare) gel filtration column. Proteins were eluted at 0.5 mL/min with 10 mM HEPES, pH 7.4, 150 mM NaCl, and appropriate fractions were pooled and concentrated.

### SPR binding experiments

SPR binding experiments were performed using a Biacore T100 biosensor equipped with a Series S CM4 sensor chip. Human DMRT, DMD-4 and its L31R mutant were immobilized over individual flow cells using amine-coupling chemistry in 10 mM HEPES-OH, pH 7.4, 150 mM NaCl, 0.005% Tween-20 at 32°C using a flow rate of 20 μL/min. Dextran surfaces were activated for 7 min using equal volumes of 0.1 M NHS(N-Hydroxysuccinimide) and 0.4 M EDC(1-Ethyl-3-(3-dimethylaminopropyl)carbodiimide) for 10 min. Each protein of interest was immobilized at 50 μg/mL in 10 mM sodium acetate, pH 4.25 for 7 min each. The immobilized surface was blocked using a 3 min injection of 1.0 M

ethanolamine, pH 8.5 resulting in immobilizations at ~3,000 RU. An unmodified surface was used as a reference flow cell to subtract bulk shift refractive index changes.

Binding experiments were performed at 25℃ in a running buffer containing 10 mM Tris-Cl pH 7.4, 150 mM NaCl, and 0.01% (v/v) Tween-20. Ubiquitin was prepared in the same buffer at a stock concentration of 300 µM and serially diluted using a three-fold dilution series from 300 to 0.137 µM. Ubiquitin analyte samples were tested in duplicate in order of increasing concentration. In each binding cycle, analytes were injected over the three immobilized surfaces at 50 µL/min for 20 s, followed by 180 s of dissociation phase, a running buffer wash step and a buffer injection at 100 µL/min for 60 s. Buffer was used instead of an analyte sample every two binding cycles to double reference the binding responses by removing systematic noise and instrument drift. The binding signal between 15 and 19 s was fit against the ubiquitin concentration using a 1:1 interaction model to calculate the $K_D$. The data was processed and analyzed using Scrubber 2.0 (BioLogic Software).

### SEC-MALS

Size exclusion chromatography with multi-angle static light scattering (SEC-MALS) was performed using an AKTA FPLC system with a Superdex 200 Increase 10/300 GL column (GE Healthcare), equilibrated with 10 mM HEPES, pH 7.4, 150 mM NaCl, with a flow rate of 0.5 mL/min. Each protein was injected at a concentration of 3 mg/mL, using an injection volume of 100 µL. UV absorbance was monitored at 280 nm using the AKTA UV detector and light-scattering and refractive index data for molecular weight determination was collected with an Optilab T-rEX and a DAWN Heleos-II detector (Wyatt Technology). Molecular weights were calculated using the Astra 6.1 software (Wyatt Technologies) using a Zimm-plot.

## Quantification of intestinal stuffing

To assay the degree of anterior intestinal stuffing in the *dmd-4* mutants *dmd-4(ot935)*, *dmd-4 (ot957ot935)*, *dmd-4(ot990ot935)*, and *dmd-4(ot933)* relative to the *him-5(e1490)* control and the positive control *ceh-34(n4796)* (which is required for pharyngeal development and was used as a positive control [*Hirose et al., 2010*]), embryos synchronized by hypochlorite treatment were plated onto NGM plates seeded with an *E. coli* OP50 strain that contains the GFP-expressing plasmid pFPV25.1 (OP50-GFP, CGC). Worms were grown to early adulthood (72 hr after egg prep, at 20℃). In order to reduce background signal, worms were washed twice in 15 mL of M9 and centrifuged at 300 x g for 1 min to separate and discard any fluorescent bacteria in suspension. Worms were then run through the Union Biometrica COPAS FP-250 Biosorter system to measure GFP signal as a function of worm length. Fluorescence data was acquired and processed with the Union Biometrica Flow-Pilot software (version 1.5.9.2). Events detected by the COPAS system were filtered attending to the time of flight (i.e. worm length) and extinction (i.e. worm opacity) to include only adult males and young adult hermaphrodites in the data set and exclude gravid hermaphrodites, as the condition of gravidity produces its own genotype-independent stuffed phenotype. Any data point without an associated parabolic time-of-flight trace was also excluded. A non-parabolic trace indicates a bent worm in the flow cell, which produces skewed fluorescence data. The green fluorescence trace measures green signal as a function of length, and the area under this curve corresponds to the total amount of fluorescence from the OP50-GFP in the anterior intestine, as confirmed by 63x-objective epifluorescent imaging (Zeiss Axio Imager.Z2). To normalize data, all data points were divided by the mean of the control group (*him-5(e1490)*).

## Statistics

Plots for expression data were generated in R using the beeswarm and ggplot2 packages. Statistical tests were performed in R as detailed in figure legends, in addition to post-hoc Bonferroni corrections to adjust p-values for number of pairwise tests in all cases where more than two pairwise statistical tests were performed.

## Acknowledgements

We thank Chi Chen for generating transgenic strains, the CGC (supported by the NIH P40 OD010440) for strains, and members of the Hobert laboratory for comments on the manuscript. This

research was supported by NIH R37 NS039996, an NRSA Kirschstein predoctoral fellowship to Emily Bayer (F31NS096863) and the Howard Hughes Medical Institute.

## Additional information

### Competing interests
Oliver Hobert: Reviewing Editor, eLife. The other authors declare that no competing interests exist.

### Funding

| Funder | Grant reference number | Author |
|---|---|---|
| Howard Hughes Medical Institute | | Oliver Hobert |
| National Institutes of Health | F31NS096863 | Emily A Bayer |
| National Institutes of Health | R37 NS039996 | Oliver Hobert |

The funders had no role in study design, data collection and interpretation, or the decision to submit the work for publication.

### Author contributions
Emily A Bayer, Conceptualization, Data curation, Formal analysis, Investigation, Writing - original draft, Writing - review and editing; Rebecca C Stecky, Lauren Neal, Phinikoula S Katsamba, Goran Ahlsen, Vishnu Balaji, Formal analysis, Investigation; Thorsten Hoppe, Conceptualization, Supervision, Writing - review and editing; Lawrence Shapiro, Conceptualization, Supervision, Funding acquisition, Writing - review and editing; Meital Oren-Suissa, Conceptualization, Formal analysis, Investigation; Oliver Hobert, Conceptualization, Supervision, Funding acquisition, Writing - original draft, Project administration, Writing - review and editing

### Author ORCIDs
Emily A Bayer (iD) http://orcid.org/0000-0002-5254-9199
Lawrence Shapiro (iD) http://orcid.org/0000-0001-9943-8819
Oliver Hobert (iD) https://orcid.org/0000-0002-7634-2854

### Decision letter and Author response
Decision letter https://doi.org/10.7554/eLife.59614.sa1
Author response https://doi.org/10.7554/eLife.59614.sa2

## Additional files

### Supplementary files
• Transparent reporting form

### Data availability
All strains are listed in the Key Resources Table and are either deposited at the CGC or available upon request.

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
