## [Decision Letter]

**Acceptance summary:**

Your paper makes a significant contribution to our understanding of how sexual dimorphism in neurons is acquired during development and how these sex-specific neurons function. The role of the genes in the DMRT/*Dsx* family and their involvement in the process are also of great importance.

Your findings document how multiple pathways intersect at different regulatory levels/mechanisms to effect a specific tissue-, sex- and temporal fate. This mechanism uses a transcription factor that has a different non-sex-specific role in a different tissue at a different time as well. This is therefore a wonderful example for how differences in regulation within the same organism make a huge difference in developmental/fate outcome. The paper is also significant because it provides important biochemical as well as developmental functionality to a previously uncharacterized, highly conserved domain of DMRT genes, which should provide insights into its vertebrate ortholog(s).

**Decision letter after peer review:**

Thank you for submitting your article "Ubiquitin-dependent regulation of a conserved DMRT protein controls sexually dimorphic synaptic pruning and behavior" for consideration by *eLife*. Your article has been reviewed by three peer reviewers, and the evaluation has been overseen by a Reviewing Editor and Piali Sengupta as the Senior Editor. The reviewers have opted to remain anonymous.

The reviewers have discussed the reviews with one another and the Reviewing Editor has drafted this decision to help you prepare a revised submission.

Your paper makes a significant contribution to our understanding of how neuronal sexual dimorphism is acquired during development and how these sex-specific neurons function. The role of the genes in the DMRT/*Dsx* family and their involvement in the process are also of great importance.

The reviewers are therefore very positive about the paper but requested that you add two points to your manuscript, data that are hopefully already available and will not require going back to doing new experiments.

– You must show the fluorescence images and control experiments for the GRASP data in Figure 3A. This experiment is very important to show synaptic remodeling.

– In order to connect the two parts of the story, additional behavioral and synaptic connectivity studies using the mutant strain with the L31R allele would strengthen the story, especially if there are novel phenotypes.

Reviewer #1:

This well-written, succinct yet complete and data-rich paper reports a novel form of regulation of a DM-domain transcription factor, DMD-4, that specifies a time- and sex-specific synaptic connectivity fate for a specific set of neurons, PHA and PHB, in *C. elegans*. In contrast to the three other known factors that specify male-specific fates, DMD-4 is found to specify female (hermaphrodite) fate and is required to do so at the larval-to-adult transition. In this respect, it is not surprising that the authors find that both the sex specification and temporal (heterochronic) specification pathways impinge (apparently as an "AND" gate) on DMD-4 promotion of sex-dimorphic neural fate. Much more surprising is the finding that a previously uncharacterized domain, the DMA domain of DMD-3, is required for the sexually dimorphic regulation of DMD-4. This domain is highly similar (authors say "homologous") to the CUE non-covalent-ubiquitin-binding domain. The authors test ubiquitin-binding and specifically the effect of residues expected to affect ubiquitin association, and find surprisingly that the domain is required for stabilizing DMD-4. The sexual dimorphism of DMD-4 stability must thus have something to do with male-specific interference of this stabilization. Also, different tissues, like the pharynx, appear to have different requirements for the ubiquitin-promoted stability (e.g. not needed for DMD-4 function in the alimentary system).

The findings are significant because they document how multiple pathways intersect at different regulatory levels/mechanisms to effect a specific tissue-, sex- and temporal fate. This mechanism uses a transcription factor that has a different non-sex-specific role in a different tissue at a different time as well. This is therefore a wonderful example for how differences in regulation within the same organism make a huge difference in developmental/fate outcome. The paper is also significant because it provides important biochemical as well as developmental functionality to a previously uncharacterized, highly conserved domain of DMRT genes. Thus, it is predictive about the functionality of this domain in vertebrates. DMD-4 itself has not been previously characterized in *C. elegans*, and provides yet another example of sexually dimorphic function for a highly conserved DMRT gene.

Reviewer #2:

This paper presents novel findings regarding how sexual dimorphism is specified in the *C. elegans* nervous system. The DMRT family has a conserved role in sex-specific cell fate identity, though the mechanisms used by this family are diverse. The novel finding is the dmd-4 has non-sex-specific expression in shared neurons early, but as animals become sexually mature, this expression is lost in males. The mechanism for protein loss is through protein degradation via a ubiquitin binding domain, another novel finding. Overall, the experimental approach is well designed, rigorous and the data support the main conclusions of the paper. More detail about image quantification methodology is important for reproducibility.

This paper makes novel and important contributions to our understanding of how neurons shared between the sexes acquire sex-specific functionality and the mechanisms that direct the dimorphism. Novel mechanistic insights about DMRT family members at the biochemical/functional level are also an important contribution.

Reviewer #3:

In this manuscript, Bayer et al. characterized a *DSX* like protein in generating the sexually dimorphic behaviors in *C. elegans*. They also identify a post-transcriptional mechanism to establish sexually dimorphic expression of this factor, DMD-4 which involves monoubiquitination. This paper adds The authors make the following claims:

The novelty of this manuscript is twofold: Firstly, the authors demonstrate a unique role for the expression of a sexually dimorphic gene in that it is expressed throughout development and later downregulated in a sex-specific manner. Secondly, the authors demonstrate that this downregulation of DMD-4 in male phasmid neurons is from the inability of the protein to be stabilized through monoubiquitination. The genetic, cell biological, and behavioral approaches support each other and is consistent with their proposed model. However, the authors fail to connect their biochemical approach identifying DMD-4 monoubiquitination with their findings in terms of neuronal connectivity and animal behavior. Listed below are some experiments that could further strengthen and expand upon their model.

1) The GRASP data in Figure 3A are important to support of the model of synaptic remodeling. Fluorescence images and control experiments should be shown.

2) An important question is whether the L31R mutant affects synaptic connectivity and behavior. This experiment would connect the two parts of the story organically.

---

## [Author Response]

[…] The reviewers are therefore very positive about the paper but requested that you add two points to your manuscript, data that are hopefully already available and will not require going back to doing new experiments.– You must show the fluorescence images and control experiments for the GRASP data in Figure 3A. This experiment is very important to show synaptic remodeling.

The wiring diagram for the data in Figure 3A is based on the data quantified in Figure 3B and representative images shown in Figure 3—figure supplement 2. We now explicitly reference these two data sources in the Figure 3A legend, in addition to the Figure 3B legend.

– In order to connect the two parts of the story, additional behavioral and synaptic connectivity studies using the mutant strain with the L31R allele would strengthen the story, especially if there are novel phenotypes.

We have now analyzed PHB>AVG connectivity in the L31R allele and found that it is phenotypically identical to the dmd-4 null allele (ot957). Based on the fact that the nervous system never expresses DMD-4 in the L31R allele, as in the dmd-4 null allele, this result was expected. These data are now shown in Figure 7—figure supplement 2A.

Reviewer #3:In this manuscript, Bayer et al. characterized a DSX like protein in generating the sexually dimorphic behaviors in *C. elegans*. They also identify a post-transcriptional mechanism to establish sexually dimorphic expression of this factor, DMD-4 which involves monoubiquitination. This paper adds The authors make the following claims:The novelty of this manuscript is twofold: Firstly, the authors demonstrate a unique role for the expression of a sexually dimorphic gene in that it is expressed throughout development and later downregulated in a sex-specific manner. Secondly, the authors demonstrate that this downregulation of DMD-4 in male phasmid neurons is from the inability of the protein to be stabilized through monoubiquitination. The genetic, cell biological, and behavioral approaches support each other and is consistent with their proposed model. However, the authors fail to connect their biochemical approach identifying DMD-4 monoubiquitination with their findings in terms of neuronal connectivity and animal behavior. Listed below are some experiments that could further strengthen and expand upon their model.1) The GRASP data in Figure 3A are important to support of the model of synaptic remodeling. Fluorescence images and control experiments should be shown.

The wiring diagram for the data in Figure 3A is based on the data quantified in Figure 3B and representative images shown in Figure 3—figure supplement 2. We now explicitly reference these two data sources in the Figure 3A legend, in addition to the Figure 3B legend.

2) An important question is whether the L31R mutant affects synaptic connectivity and behavior. This experiment would connect the two parts of the story organically.

We have now analyzed PHB>AVG connectivity in the L31R allele and found that it is phenotypically identical to the dmd-4 null allele (ot957). Based on the fact that the nervous system never expresses DMD-4 in the L31R allele, as in the dmd-4 null allele, this result was expected – and it now nicely connects the biochemical approach to a cellular read-out. These data are now shown in Figure 7—figure supplement 2A.